# A modeling study of the nonlinear response of fine particles to air pollutant emissions in the Beijing-Tianjin-Hebei region

**Bin Zhao[1,2,3], Wenjing Wu[1,2], Shuxiao Wang[1,2], Jia Xing[1,2], Xing Chang[1,2], Kuo-Nan Liou[3], Jonathan H. Jiang[4], Yu Gu[3], Carey Jang[5], Joshua S. Fu[6], Yun Zhu[7], Jiandong Wang[1,2], Jiming Hao[1,2]**

[1] School of Environment, and State Key Joint Laboratory of Environment Simulation and Pollution Control, Tsinghua University, Beijing 100084, China

[2] State Environmental Protection Key Laboratory of Sources and Control of Air Pollution Complex, Beijing 100084, China

[3] Joint Institute for Regional Earth System Science and Engineering and Department of Atmospheric and Oceanic Sciences, University of California, Los Angeles, CA 90095, USA

[4] Jet propulsion Laboratory, California Institute of Technology, Pasadena, CA 91109, USA

[5] U.S. Environmental Protection Agency, Research Triangle Park, NC 27711, USA

[6] Department of Civil and Environmental Engineering, University of Tennessee, Knoxville, TN 37996, United States

[7] School of Environmental Science and Engineering, South China University of Technology, Guangzhou 510006, China

Correspondence to: Shuxiao Wang (shxwang@tsinghua.edu.cn)

**Abstract.**

The Beijing-Tianjin-Hebei (BTH) region has been suffering from the most severe fine particle ($PM_{2.5}$) pollution in China, which causes serious health damage and economic loss. Quantifying the source contributions to $PM_{2.5}$ concentrations has been a challenging task because of the complicated non-linear relationships between $PM_{2.5}$ concentrations and emissions of multiple pollutants from multiple spatial regions and economic sectors. In this study, we use the Extended Response Surface Modeling (ERSM) technique to investigate the

nonlinear response of $PM_{2.5}$ concentrations to emissions of multiple pollutants from different
regions and sectors over the BTH region, based on over 1000 simulations by a chemical
transport model (CTM). The ERSM-predicted $PM_{2.5}$ concentrations agree well with
independent CTM simulations, with correlation coefficients larger than 0.99 and mean
normalized errors less than 1%. Using the ERSM technique, we find that, among all air
pollutants, primary inorganic $PM_{2.5}$ makes the largest contribution (24-36%) to $PM_{2.5}$
concentrations. The contribution of primary inorganic $PM_{2.5}$ emissions is especially high in
heavily polluted winter, and is dominated by the industry as well as residential and
commercial sectors, which should be prioritized in $PM_{2.5}$ control strategies. The total
contributions of all precursors (nitrogen oxides, $NO_X$; sulfur dioxides, $SO_2$; ammonia, $NH_3$;
non-methane volatile organic compounds, NMVOC; intermediate-volatility organic
compounds, IVOC; primary organic aerosol, POA) to $PM_{2.5}$ concentrations range between 31%
and 48%. Among these precursors, $PM_{2.5}$ concentrations are primarily sensitive to the
emissions of $NH_3$, NMVOC+IVOC, and POA. The sensitivities increase substantially for $NH_3$
and $NO_X$, and decrease slightly for POA and NMVOC+IVOC with the increase in the
emission reduction ratio, which illustrates the nonlinear relationships between precursor
emissions and $PM_{2.5}$ concentrations. The contributions of primary inorganic $PM_{2.5}$ emissions
to $PM_{2.5}$ concentrations are dominated by local emission sources, which account for over 75%
of the total primary inorganic $PM_{2.5}$ contributions. For precursors, however, emissions from
other regions could play similar roles as local emission sources in the summer and over the
northern part of BTH. The source contribution features for various types of heavy-pollution
episodes are distinctly different from each other, and from the monthly mean results,
illustrating that control strategies should be differentiated based on the major contributing
sources during different types of episodes.

## 1   Introduction

China is one of the regions with highest concentration of $PM_{2.5}$ (particulate matter with
aerodynamic diameter equal to or less than 2.5 μm) in the world (van Donkelaar et al., 2015).
The problem is especially serious over the Beijing-Tianjin-Hebei (BTH) region, one of the
most populous and developed regions in China. Annual average $PM_{2.5}$ concentrations in this
region reached 85-110 μg/m$^3$ during 2013-2015, which approximately triple the standard
threshold (35 μg/m$^3$) and far exceed those in other metropolitan regions (Wang et al., 2017). It
has been estimated that the severe $PM_{2.5}$ pollution leads to about 1.05-1.23 million premature
deaths per year in China (Lim et al., 2012; Burnett et al., 2014; Wang et al., 2016b), and the
monetized loss over the BTH region is as high as 134 billion Chinese Yuan, representing 2.2%
of regional gross domestic product (GDP) (Lv and Li, 2016). Additionally, $PM_{2.5}$ substantially
affects global and regional climate by absorbing and scattering solar radiation and by altering
cloud properties (IPCC, 2013; Seinfeld et al., 2016; Zhao et al., 2017a), which in turn exert
impact on regional air quality (Wang et al., 2014a; Zhao et al., 2017b).
To tackle the heavy $PM_{2.5}$ pollution problem, Chinese government issued the "Action Plan
on Prevention and Control of Air Pollution" in September 2013, which aimed at a 25%
reduction in $PM_{2.5}$ concentrations over the BTH region by 2017 from the 2012 levels (The
State Council of the People's Republic of China, 2013). The attainment of ambient $PM_{2.5}$
standard would further require substantial reductions in air pollutant emissions (Wang et al.,
2017; Wang et al., 2015). To establish emission control strategies, many studies have
apportioned the sources of $PM_{2.5}$ over the BTH region, either by mining monitoring data using
the Positive Matrix Factorization and Chemical Mass Balance methods (e.g., Zhang et al.,
2007; Yu et al., 2013) or by embedding chemical tracers in chemical transport models (CTMs)
(e.g., Wang et al., 2016c; Li et al., 2015b; Ying et al., 2014). While these studies can capture
the current contributions of various sources to $PM_{2.5}$ concentrations, these contributions could
differ significantly from the $PM_{2.5}$ reductions induced by reducing emissions from the
corresponding sources, due to highly nonlinear chemical mechanisms (Han et al., 2016; Wang
et al., 2011). Therefore, it is imperative to assess the nonlinear response of $PM_{2.5}$ to pollutant
emissions from multiple sources, which could provide direct support for the development of
effective control policies.
The most widely used technique to evaluate the responses of $PM_{2.5}$ concentrations to
emission changes is the "Brute force" method, which involves perturbing emissions from a
certain source and repeated solution of a CTM (Russell et al., 1995). A number of studies
have utilized the "Brute force" method to quantify the sensitivities of $PM_{2.5}$ concentrations
over the BTH region to emissions from different spatial regions (Streets et al., 2007; Wang et
al., 2008; Li and Han, 2016; Wang et al., 2014b) or different economic sectors (Wang et al.,
2008; Han et al., 2016; Wang et al., 2014b; Liu et al., 2016), either on a seasonal basis
(Streets et al., 2007; Wang et al., 2008; Han et al., 2016; Liu et al., 2016) or during a specific
heavy-pollution episode (Li and Han, 2016; Wang et al., 2014b). To improve the

computational efficiency, several mathematic techniques embedded in CTMs have been developed to simultaneously calculate the sensitivities of the modeled concentrations to multiple emission sources, including the Decoupled Direct Method (Yang et al., 1997) and Adjoint Analysis (Sandu et al., 2005; Hakami et al., 2006). Zhang et al. (2016) used the Adjoint Analysis method to examine sensitivities of $PM_{2.5}$ concentrations in the BTH region to pollutant emissions during several pollution periods. However, all the preceding studies only quantified first-order sensitivities and therefore could not capture the nonlinearity in the responses of $PM_{2.5}$ concentrations to pollutant emissions, which can be extremely strong in metropolitan regions like BTH due to complex chemical mechanisms (Wang et al., 2011). Moreover, no studies have simultaneously evaluated the response of $PM_{2.5}$ concentrations in BTH to emissions of multiple pollutants from different sectors and regions, which we need to consider and balance to develop cost-effective control strategies.

In light of the drawbacks of the preceding methods, the Response Surface Modeling (RSM) technique (denoted by "conventional RSM" technique hereafter to distinguish from the ERSM technique) has been developed by using advanced statistical techniques to characterize the complex nonlinear relationship between model outputs and inputs (U.S. Environmental Protection Agency, 2006; Xing et al., 2011; Wang et al., 2011). This technique has been applied to the United States (U.S. Environmental Protection Agency, 2006) and the Eastern China (Wang et al., 2011) to evaluate the response of $PM_{2.5}$ and its chemical components to pollutant emissions. However, the number of emission scenarios required to build conventional RSM depends on the variable number via an equation of fourth or higher order (Zhao et al., 2015b). Therefore, the required scenario number would be tens of thousands for over 15 variables and even hundreds of thousands for over 25 variables, which is computationally impossible for most three-dimensional CTMs. To overcome this major limitation, we recently developed the Extended Response Surface Modeling (ERSM) technique (Zhao et al., 2015b), which substantially reduced the scenario number needed to build the response surface and hence extended its applicability to an increased number of regions, pollutants, and sectors with an acceptable computational burden.

Given the advantage of the ERSM technique, here we apply it to over 1000 simulations by the Community Multi-scale Air Quality model with Two-Dimensional Volatility Basis Set (CMAQ/2D-VBS) to systematically evaluate the nonlinear response of $PM_{2.5}$ to emission changes of multiple pollutants from different sectors and regions over the BTH region. The

major sources contributing to $PM_{2.5}$ and its major components are identified and the nonlinearity in the response of $PM_{2.5}$ to emission changes is characterized. Based on results of this study, suggestions for $PM_{2.5}$ control policies over the BTH region are proposed.

## 2 Methodology

### 2.1 CMAQ/2D-VBS configuration and evaluation

The CMAQ/2D-VBS model was developed in our previous study (Zhao et al., 2016) by incorporating the 2D-VBS model framework into CMAQv5.0.1. Compared with the default CMAQ, the CMAQ/2D-VBS model explicitly simulates aging of secondary organic aerosol (SOA) formed from non-methane volatile organic compounds (NMVOC), aging of primary organic aerosol (POA), and photo-oxidation of intermediate-volatility organic compounds (IVOC), thereby significantly improving the simulation results of organic aerosol (OA), particularly SOA. The model parameters within the 2D-VBS framework have been optimized in our previous studies (Zhao et al., 2015a; Zhao et al., 2016) based on a series of smog-chamber experiments. Here we use the same model parameters as those of the "High-Yield VBS" configuration reported in Zhao et al. (2016), which agrees best with surface OA and SOA observations among three model configurations. An application in the Eastern China reveals that CMAQ/2D-VBS reduces the underestimation in OA concentrations from 45% (default CMAQv5.0.1) to 19%. More importantly, while the default CMAQv5.0.1 substantially underestimates the fraction of SOA in OA by 5–10 times and can not track oxygen-to-carbon ratio (O:C), the SOA fraction and O:C simulated by CMAQ/2D-VBS agree fairly well with observations.

We apply the CMAQ/2D-VBS model over the BTH region. One-way, double nesting simulation domains are used, as shown in Fig. 1. Domain 1 covers East Asia with a grid resolution of 36 km×36 km; domain 2 covers the BTH and its surrounding regions with a grid resolution of 12 km×12 km. We use the SAPRC99 gas-phase chemistry module and the AERO6 aerosol module, in which the treatment of OA is replaced with the 2D-VBS framework. The aerosol thermaldynamics is based on ISORROPIA-II. The initial and boundary conditions for Domain 1 are kept constant as the model default profile, and those for Domain 2 are extracted from the output of Domain 1. A 5-day spin-up period is used to reduce the influence of initial conditions on modeling results.

The Weather Research and Forecasting Model (WRF, version 3.7) is used to generate the meteorological fields. The National Center for Environmental Prediction (NCEP)'s FNL

(Final) Operational Global Analysis data (ds083.2) at 1.0º × 1.0º and 6-h resolution are used
to generate the first guess field. The NCEP's Automated Data Processing (ADP) data
(ds351.0 and ds461.0) are used in objective analysis (i.e., grid nudging). The major physics
options for WRF include the Kain-Fritsch cumulus scheme, the Pleim-Xiu land-surface
module, the Asymmetric Convective Model with non-local upward mixing and local
downward mixing (ACM2) for planetary boundary layer (PBL) parameterization, the
Morrison double-moment scheme for cloud microphysics, and the Rapid Radiative Transfer
Model for GCMs (RRTMG) radiation scheme. The land cover type data are obtained from the
Moderate resolution Imaging Spectroradiometer (MODIS). The simulation periods are
January, March, July, and October in 2014, representing winter, spring, summer, and fall. We
select these four months because the occurrence frequencies of various meteorological types
in these months are statistically most similar to the average conditions in winter, spring,
summer, and fall during 2004-2013 (Wu, 2016).
A high-resolution anthropogenic emission inventory in 2014 has been developed using an
"emission factor method" (Fu et al., 2013; Zhao et al., 2013b) for the BTH region by
Tsinghua University. The emissions from area and mobile sources are first calculated for each
prefecture-level city based on statistical data, and subsequently distributed into the model
grids according to spatial distribution of population, GDP, and road networks. A unit-based
method (Zhao et al., 2008) is applied to estimate and locate the emissions from large point
sources (LPS) including power plants, iron and steel plants, and cement plants. The
anthropogenic emission inventory in other provinces of China was originally developed for
2010 and 2012 in our previous studies (Zhao et al., 2013b; Zhao et al., 2013a; Wang et al.,
2014c; Cai et al., 2016), which has been updated to 2014 in this study following the same
methodology. In both the BTH and national emission inventories, the emissions from open
burning of agricultural residue are calculated using crop yields, straw to grain ratio, fraction
of biomass burned in the open field, and emission factors (Fu et al., 2013; Zhao et al., 2013b;
Wang and Zhang, 2008). We do not include the emissions from forest and grassland fires,
which typically account for less than 5% of the total biomass burning emissions over the BTH
region (Qin and Xie, 2011) and are not the focus of the present study. Table S1 summarizes
emissions of major air pollutants in each prefecture-level city over the BTH region in 2014;
Table S2 gives the provincial emissions in the whole China in 2014. The emissions for other
countries are obtained from the MIX emission inventory (Li et al., 2015a) for 2010, which is

the latest year available. Following our previous study (Zhao et al., 2016), we assume IVOC emissions to be 30 times, 4.5 times, 1.5 times, and 3.0 times the POA emissions from gasoline vehicles, diesel vehicles, biomass burning, and other emission sources, respectively, which is based on a series of laboratory measurements (Gordon et al., 2014b; Gordon et al., 2014a; Hennigan et al., 2011; Jathar et al., 2014). The biogenic emissions were calculated by the Model of Emissions of Gases and Aerosols from Nature (MEGAN; Guenther et al., 2006).

We compared the simulation results of WRFv3.7 and CMAQ/2D-VBS with meteorological observations obtained from the National Climatic Data Center (NCDC), $PM_{2.5}$ observations at 138 state-controlled observational sites, and observations of major $PM_{2.5}$ chemical components at 7 sites within the modeling domain. We show that the meteorological and chemical simulations generally agree well with observations, with performance statistics mostly within the benchmark values proposed by previous studies. Details of the model evaluation methods and results are given in the Supplementary Information (Section 1, Table S3-S5, Fig. S1-S5).

## 2.2  Development of ERSM prediction system

The detailed methodologies of the conventional RSM and ERSM techniques have been described in our previous papers (Zhao et al., 2015b; Xing et al., 2011). Here we only summarize some key components. The conventional RSM technique characterizes the relationships between a response variable (e.g., $PM_{2.5}$ concentration) and a set of control variables (i.e., emissions of particular pollutants from particular sources) based on a number of randomly generated emission control scenarios (Xing et al., 2011; Wang et al., 2011). The $PM_{2.5}$ concentration for each emission scenario is calculated with a CTM (CMAQ/2D-VBS in this study), and the conventional RSM is subsequently established using the Maximum Likelihood Estimation - Empirical Best Linear Unbiased Predictors (MLE-EBLUPs) developed by Santner et al. (2003). Due to the limitation of the conventional RSM technique with respect to variable number, we have developed the ERSM technique (Zhao et al., 2015b) to extend the applicability to an increased number of variables and geographical regions. The ERSM technique first quantifies the relationship between $PM_{2.5}$ concentrations and precursor emissions for each single region using the conventional RSM technique as described above, and then assesses the effects of inter-regional transport of $PM_{2.5}$ and its precursors on $PM_{2.5}$ concentration in the target region. In order to quantify the interaction among regions, we introduce a key assumption that the emissions of precursors in the source region affect $PM_{2.5}$

concentrations in the target region through two major processes: (1) the inter-regional
transport of precursors enhancing the chemical formation of secondary $PM_{2.5}$ in the target
region; (2) the formation of secondary $PM_{2.5}$ in the source region followed by transport to the
target region. We quantify the individual contributions of these two processes as well as the
contribution of local emissions in the target region, which are subsequently integrated to
derive the total $PM_{2.5}$ concentrations in the target region. The development of the ERSM
prediction system requires several hundred to over 1000 emission scenarios, but once built, it
enables real-time prediction of $PM_{2.5}$ concentrations for any given control strategy and proves
to be an efficient and user-friendly decision making tool. Moreover, ERSM can be applied to
design least-cost control strategy once it is coupled with control cost models/functions that
links the emission reductions with economic costs.
For application of the RSM/ERSM techniques to the BTH region, we define 5 target
regions in the inner modeling domain (Domain 2), i.e., Beijing, Tianjin, Northern Hebei (N
Hebei), Eastern Hebei (E Hebei), and Southern Hebei (S Hebei), as shown in Fig. 1. The
decomposition of the Hebei province is based on a preliminary analysis of the pollutant
transport patterns over the BTH region (Section 2 in the Supplementary Information). The
simulation using back trajectory method indicates that four major types of heavy-pollution
episodes in Beijing are primarily contributed by air mass from the south, the local area, the
northwest, and the southeast. We develop two RSM/ERSM prediction systems (Table 1). The
response variables for the first prediction system, which is built using the conventional RSM
technique, are concentrations of $PM_{2.5}$, $SO_4^{2-}$, $NO_3^-$, and OA over the urban areas of
prefecture-level cities in the five target regions. For the second prediction system that is
established using the ERSM technique, the response variables are only $PM_{2.5}$ concentrations.
The first prediction system use 101 emission control scenarios generated by the Latin
Hypercube Sample (LHS) method (Iman et al., 1980) to map atmospheric concentrations
versus emissions of five $PM_{2.5}$ precursors, i.e., $NO_X$, $SO_2$, $NH_3$, NMVOC+IVOC, and POA, in
all five target regions (Table 1). It is on one hand intended for the validation of the second
system (Section 3.1), and on the other hand used to study the source contributions of major
$PM_{2.5}$ components. For the second system, the emissions of the preceding $PM_{2.5}$ precursors as
well as primary inorganic $PM_{2.5}$ (i.e., the chemical components of primary $PM_{2.5}$ other than
POA) in each of the 5 regions are categorized into 7 and 4 control variables, respectively,
resulting in 55 control variables in total (Table 1). Note that we distinguish POA and primary
inorganic PM$_{2.5}$ because the former undergoes chemical reactions and produces SOA, while
the latter is mostly chemically inert in the CMAQ/2D-VBS model. We generate 1121
scenarios (see Table 1) to build the response surface, following the method detailed in Zhao et
al. (2015b). Specifically, the scenarios include (1) 1 CMAQ/2D-VBS base case; (2) 200
scenarios generated by applying LHS method for the control variables of precursors in
Beijing, 200×4 scenarios generated in the same way for Tianjin, Northern Hebei, Eastern
Hebei, and Southern Hebei; (3) 100 scenarios generated by applying LHS method for the total
emissions of NO$_X$, SO$_2$, NH$_3$, NMVOC+IVOC, and POA in all 5 regions; and (4) 20
scenarios where one of the control variables of primary inorganic PM$_{2.5}$ emissions is set to
0.25 for each scenario. Here the scenario numbers (200 in group 2 and 100 in group 3) are
determined based on numerical experiments conducted in our previous studies (Xing et al.,
2011; Wang et al., 2011), which showed that the response surface for 7 and 5 variables could
be built with good prediction performance (mean normalized error < 1%; correlation
coefficient > 0.99) using 200 and 100 scenarios, respectively. Finally, we generate 54
independent scenarios for out-of-sample validation, which will be detailed in Section. 3.1.
For application of the ERSM prediction system to quantitatively characterize the
sensitivity of PM$_{2.5}$ concentrations to emission changes, we define "PM$_{2.5}$ sensitivity" as the
change ratio of PM$_{2.5}$ concentration divided by the reduction ratio of a emission source,
following previous studies (Zhao et al., 2015b; Wang et al., 2011).
$$S_a^X = \left[\left(C^* - C_a\right)/C^*\right]/\left(1-a\right) \qquad (4)$$
where $S_a^X$ is the PM$_{2.5}$ sensitivity to emission source $X$ at its emission ratio $a$; $C^*$ and $C_a$ are
PM$_{2.5}$ concentrations in the base case (when the emission ratio of $X$ is 1) and in the control
scenario where the emission ratio of $X$ is $a$, respectively. Similar indices can be defined for
chemical components of PM$_{2.5}$, such as NO$_3^-$, SO$_4^{2-}$, and OA.

## 3   Results and discussion

### 3.1   Validation of ERSM performance

The conventional RSM technique has been extensively demonstrated to have high accuracy
and stability in previous papers (Xing et al., 2011; Wang et al., 2011), so we only describe the
validation of the ERSM technique. Following Zhao et al. (2015b), we assess the performance

of the ERSM prediction system using the "out-of-sample" and 2D-isopleths validation methods, which focus on the accuracy and stability of the prediction system, respectively.

For out-of-sample validation, we use the ERSM prediction system to calculate the $PM_{2.5}$ concentrations for 54 "out-of-sample" control scenarios, i.e., scenarios independent from those used to build the prediciton system, and compare with the corresponding CMAQ/2D-VBS simulation results. These 54 out-of-sample scenarios (summarized in Table S6) include 40 cases (case 1-40) where the control variables of precursors change but those of primary inorganic $PM_{2.5}$ stay the same as the base case, 4 cases (case 41-44) the other way around, and 10 cases (case 45-54) where control variables of precursors and primary inorganic $PM_{2.5}$ change simultaneously. Most cases are generated randomly with the LHS method (case 4-6, 10-12, 16-18, 22-24, 28-54), and some cases are designed where all control variables are subject to large emission changes (case 1-3, 7-9, 13-15, 19-21, 25-27).

Figure 2 compares the ERSM-predicted and CMAQ/2D-VBS-simulated $PM_{2.5}$ concentrations and $PM_{2.5}$ responses (defined as the difference between $PM_{2.5}$ concentration in an emission control scenario and that in the base case) for the out-of-sample scenarios using scatter plots. Table 2 summarizes the statistics of the model performance. The definitions of normalized error (NE), mean normalized error (MNE), and normalized mean error (NME) are given as follows:

$$NE = |P_i\text{-}S_i|/S_i \tag{1}$$

$$MNE = \frac{1}{Ns}\sum_{i=1}^{Ns}\left[|P_i\text{-}S_i|/S_i\right] \tag{2}$$

$$NME = \sum_{i=1}^{Ns}|P_i\text{-}S_i|/\sum_{i=1}^{Ns}S_i \tag{3}$$

where $P_i$ and $S_i$ are the ERSM-predicted and CMAQ/2D-VBS-simulated value of the i[th] out-of-sample scenario; $Ns$ is the number of out-of-sample scenarios. Figure 2 shows that the ERSM predictions and CMAQ/2D-VBS simulations agree well with each other. For $PM_{2.5}$ concentrations, the correlation coefficients are larger than 0.99, and the MNEs and NMEs are less than 1% for all four months. The maximum NEs could be as large as 11% for particular month and region, but the 95% percentiles of NEs are all within 4.4%. NEs exceeding 4.4% happen only for the scenarios where most control variables are reduced substantially, indicating relatively large errors at low emission rates, which is consistent with our previous study (Zhao et al., 2015b). Note that all sensitivity scenarios used in Sections 3.2-3.4 have $\leq$ 80% emission reductions, which helps to avoid relatively large errors. We also examine the errors in predicted $PM_{2.5}$ response. Since the CMAQ/2D-VBS-simulated $PM_{2.5}$ responses are

very close to zero in several scenarios, their normalized errors (NEs) and mean normalized errors (MNEs) could be extremely large even if the absolute errors are small, which cannot properly characterize the accuracy of the ERSM technique. For this reason, we only calculate the correlation coefficients and NMEs (Table 2). The correlation coefficients of $PM_{2.5}$ response are larger than 0.99, and the NMEs are within 5.6% for all months. In summary, the out-of-sample validation indicates an overall good agreement between ERSM predictions and CMAQ/2D-VBS simulations.

We further examine whether the ERSM technique can capture the trends in $PM_{2.5}$ concentrations in response to continuous changes in precursor emissions, i.e., the stability of the ERSM technique. To this end, we compare the 2D-isopleths of $PM_{2.5}$ concentrations as a function of simultaneous changes in two precursors' emissions in all five regions derived from the ERSM and conventional RSM techniques. It should be noted that, although the ERSM technique is applicable to a much larger number of control variables than conventional RSM, the assumptions in the treatment of inter-regional transport (Section 2.2) in ERSM might affect its accuracy. Nevertheless, the predictions by conventional RSM can be regarded as proxies for real CMAQ/2D-VBS simulations since it has been extensively demonstrated to have high accuracy and stability in previous studies (Xing et al., 2011; Wang et al., 2011). For this reason, the comparison between the ERSM and conventional RSM techniques helps to evaluate the stability of the ERSM technique. Figure 3 illustrates the $PM_{2.5}$ isopleths in Beijing as a function of three combinations of precursors, i.e., $NO_X$ vs $NH_3$, $SO_2$ vs $NH_3$, and VOC+IVOC vs POA; the isopleths for other regions are very similar and thus not shown. The X- and Y-axis of the figures represent the "emission ratio", defined as the ratios of the changed emissions to the emissions in the base case. For example, an emission ratio of 0.7 means the emission of a particular control variable accounts for 70% that of the base case. The colour isopleths represent $PM_{2.5}$ concentrations. The comparison shows that the shapes of isopleths derived from both prediction systems generally agree with each other. The agreement is very good for the case of VOC+IVOC vs POA, and for the cases of $NO_X$ vs $NH_3$ and $SO_2$ vs $NH_3$ when the emission ratios for $NO_X$ and $NH_3$ are larger than 0.2. Relatively large errors occur at very low $NO_X/NH_3$ emission ratios (< 0.2) due primarily to an extremely strong nonlinearity. Within these low emission ranges, the ERSM technique can capture the general trends in $PM_{2.5}$ concentrations in response to emission changes, but the concentration gradients predicted by ERSM are smaller than those given by conventional RSM. More

studies are needed to further improve the performance of ERSM at very low $NO_X/NH_3$
emission ratios. Despite the existing errors, the general consistency between RSM and
ERSM-predicted isopleths demonstrates the stability of the ERSM prediction system. In other
words, the discrepancies between ERSM and CMAQ/2D-VBS cannot challenge the major
conclusions on the effectiveness of emission reductions. Finally, as stated in the last
paragraph, all sensitivity scenarios used in the following discussions have emission ratios $\geq$
0.2, since < 0.2 emission reductions are quite rare as limited by the technologically feasible
reduction potentials (Wang et al., 2014c).
**3.2   Response of PM$_{2.5}$ concentrations to emissions of air pollutants**
Having demonstrated the reliability of the ERSM prediction system, we employ it to
investigate the responses of PM$_{2.5}$ concentrations to emissions of various pollutants from
different sectors and regions. We use "PM$_{2.5}$ sensitivity" defined in Section 2.2 to
quantitatively characterize the sensitivity of PM$_{2.5}$ concentrations to emission changes. Figure
4 illustrates the sensitivity of 4-month (January, March, July, and October) mean PM$_{2.5}$
concentrations to stepped control of individual air pollutants (left panel) and individual
pollutant-sector combinations (right panel) in the BTH region, which are derived from the
ERSM technique. The left panel of Fig. 4 can be obtained from both the RSM and ERSM
prediction systems and their results are consistent, whereas the right panel of Fig. 4, as well as
the results shown in Fig. 5 and 6 can only be derived from ERSM. Among all pollutants, the
4-month mean PM$_{2.5}$ concentrations are most sensitive to the emissions of primary inorganic
PM$_{2.5}$ in all five regions, and the PM$_{2.5}$ sensitivities vary from 24% to 36% according to
region. When primary inorganic PM$_{2.5}$ emissions from various sectors are differentiated, the
industry sector is found to make the largest contribution to PM$_{2.5}$ concentrations, followed by
the residential and commercial sectors; the contribution of power plants is negligibly small
because of smaller emissions and higher stacks. The PM$_{2.5}$ sensitivities to primariy inorganic
PM$_{2.5}$ emissions remain constant at various reduction ratios.
While primary inorganic PM$_{2.5}$ makes the largest contribution to PM$_{2.5}$ concentrations
among all air pollutants, the total contributions of all precursors (NO$_X$, SO$_2$, NH$_3$, NMVOC,
IVOC, and POA), which range between 31% and 48%, exceed that of primary inorganic
PM$_{2.5}$ (24-36%). Among the precursors, PM$_{2.5}$ concentrations are primarily sensitive to the
emissions of NH$_3$, NMVOC+IVOC, and POA, and their relative importance differ according
to reduction ratio. The PM$_{2.5}$ sensitivity to NH$_3$ increases substantially with the increase of

reduction ratio, primarily attributable to the transition from $NH_3$-rich to $NH_3$-poor regimes when more controls are enforced. The $PM_{2.5}$ sensitivies to POA and NMVOC+IVOC, however, decrease slightly with the increase of reduction ratio. This is because that, based on the gas-particle absorptive partitioning theory, organics have a higher tendency to partition into the particle phase at larger OA concentrations. As a result of the nonlinearity, the $PM_{2.5}$ sensitivities to POA and NMVOC+IVOC emissions are larger than those to $NH_3$ emissions at small reduction ratios (e.g., 20%), while it is the other way around at large reduction ratios (e.g., 80%).

The $PM_{2.5}$ sensitivity to $SO_2$ emissions is considerably smaller compared with the three precursors above, and does not change significantly as a function of reduction ratio. From 2007 to 2014 (the base year of this study), both $SO_2$ emissions and $SO_4^{2-}$ concentrations in $PM_{2.5}$ have been continuously decreasing due to effective control policies (Wang et al., 2017), which partly explains the small sensitivity of $PM_{2.5}$ to $SO_2$ emissions. The response of $PM_{2.5}$ concentrations to $NO_X$ emissions could change from negative to positive with the increase of reduction ratio, which has been reported in several previous studies (Dong et al., 2014; Zhao et al., 2013c; Cai et al., 2016). Small $NO_X$ emission reductions could lead to increase in $O_3$ and $HO_X$ concentrations in several seasons owing to a NMVOC-limited photochemical regime, which on one hand enhances $SO_4^{2-}$ and SOA formation, and on the other hand, could also increase $NO_3^-$ concentrations by accelerating the nocturnal formation of $N_2O_5$ and $HNO_3$ through the $NO_2 + O_3$ reaction at low temperatures. A substantial reduction in $NO_X$ emissions, however, transforms the NMVOC-limited regime to a $NO_X$-limited regime, resulting in a successive decline in concentrations of $O_3$, $HO_X$, and most $PM_{2.5}$ chemical components. Judging from the our simulation results (Fig. 4), if only the $NO_X$ emissions within the BTH region are controlled, a very large reduction ratio of about 80% is required to realize a reduction in annual $PM_{2.5}$ concentrations in most areas. However, the effects could be distinctly different if $NO_X$ emissions outside the BTH region are jointly reduced. Our previous studies using the CMAQ model (Zhao et al., 2013c; Wang et al., 2010; Wang et al., 2011) have shown that uniform reductions in $NO_X$ emissions in the whole China by 23-50% result in considerable annual $PM_{2.5}$ reduction over the BTH region. This is because $NO_X$ emission reductions in upwind regions are more likely to result in a net $PM_{2.5}$ decrease compared with local emission reductions, since the photochemistry typically changes from a NMVOC-limited regime in local urban areas at surface to a $NO_X$-limited regime in downwind

areas or at upper levels (Xing et al., 2011). The results shown in Fig. 4 also support the above-
mentioned pattern and mechanism to some extent: even a 20% $NO_X$ emission reduction in
BTH can lead to $PM_{2.5}$ decrease in Northern Hebei, because, as the northernmost region in
BTH, it is significantly affected by emissions in other regions within BTH. Note that some
recently discovered chemical pathways are missing in the model, such as the oxidation of $SO_2$
by $NO_2$ in aerosol water and the $SO_2$ heterogeneous reactions on the dust surface (Fu et al.,
2016; Cheng et al., 2016; Wang et al., 2016a). Incorporation of these processes in the model
may affect the simulated responses of $PM_{2.5}$ to $NO_X$ and $SO_2$ emissions. Regarding emission
sectors, the contributions of $SO_2$ and $NO_X$ emissions are domiated by "other sources" (sources
other than LPS) because they emit larger amount of pollutants at lower height compared with
LPS.

12       The black dotted lines in Fig. 4 show the $PM_{2.5}$ sensitivity when all pollutants from all

sectors are controlled simultaneously. The sum of $PM_{2.5}$ sensitivities to individual pollutant-
sector combinations (stacked columns) is mostly larger than the sensitivity to all pollutants
and sectors (black dotted lines), especially under large reduction ratios. This is mainly
attributed to the overlapping effect of two precursors (e.g., $SO_2$ and $NH_3$) involved in the
formation of ammonium sulfate and ammonium nitrate. Nevertheless, at small reduction
ratios, the sum of individual sensitivities is sometimes smaller, because the negative effects of
reducing $NO_X$ are mitigated when we simultaneously reduce $NO_X$ emissions from multiple
sectors as well as emissions of other air pollutants such as NMVOC. When all pollutants and
sectors are controlled together, the $PM_{2.5}$ sensitivity generally increases with reduction ratio,
indicating that additional air quality benefit could be achieved, larger than the expectation
from linear extropolation, if more control measures are implemented.

24       Figure 5 illustrates the $PM_{2.5}$ sensitivities to individual pollutant-sector combinations in

each month. The source contribution features are significantly discrepant in different months.
The contributions of primary inorganic $PM_{2.5}$ emissions to $PM_{2.5}$ concentrations are notably
higher in January than in other months, which is probably attributed to weaker dilution and
slower chemical reactions in January. Regarding different emission sectors of primary
inorganic $PM_{2.5}$, the industrial sector plays a dominant role in all months except January,
when the residential and commercial sectors make a similar or even larger contribution as
compared to the industrial sector. The higher contribution of the residential and commercial
sectors in January is on one hand because of the higher emissions due to heating, and on the
other hand explained by weaker vertical mixing in winter, which results in a larger relative
contribution of low-level sources. This result highlights the importance of residential and
commercial sources for $PM_{2.5}$ pollution controls in the winter. The contributions of precursors
are dominated by POA and NMVOC+IVOC in January, while in July, $NO_X$, $SO_2$, and $NH_3$,
which are known to be precursors of secondary inorganic aerosols, make larger contributions
than POA and NMVOC+IVOC. The responses of $PM_{2.5}$ concentrations to $NO_X$ emissions can
be opposite in different seasons. Specifically, in July, $NO_X$ emission reductions always induce
decrease in $PM_{2.5}$ concentrations due to a $NO_X$-limited photochemical regime. In January,
however, even a 80% reducion in $NO_X$ emissions (roughly the maximum technically feasible
reduction ratio) could result in a net $PM_{2.5}$ increase, as a result of a strong NMVOC-limited
regime. To achieve a net $PM_{2.5}$ reduction in January, it would be necessary to simultaneously
reduce $NO_X$ emissions outside the BTH region.

13       We further evaluate the contributions of primary inorganic $PM_{2.5}$ and precursor emissions

from various regions to $PM_{2.5}$ concentrations (Fig. 6, Fig. S6). Here the contributions are
quantified by comparing the base case with sensitivity scenarios in which emissions from a
specific source are reduced by 80%, which reaches the maximum technologically feasible
reduction ratios of major pollutants in most areas (Wang et al., 2014c). Obviously, the
contributions of total primary inorganic $PM_{2.5}$ emissions in the BTH region are dominated by
local sources, which account for over 75% of the total primary inorganic $PM_{2.5}$ contributions.
When precursor emissions are decomposed into different regions, local sources usually also
represent the largest contributors, but precursor emissions from other regions (denoted by
"regional precursor emissions" hereafter) could also make significant contributions,
depending on regions and seasons. The precursor emissions from the northern part of BTH
(e.g., Northern Hebei, Beijing) mainly contribute to local $PM_{2.5}$ concentrations, whereas those
from the southern part of BTH (e.g., Southern Hebei) significantly affect the $PM_{2.5}$
concentrations in both the local region and other regions. Over the BTH, heavy pollution is
frequently associated with southerly wind while strong northerly wind often blows away
$PM_{2.5}$ pollution (Jia et al., 2008; Zheng et al., 2015), which explains the higher contribution of
emissions from southern BTH to other regions. Moreover, the importance of regional
precursor emissions relative to local ones is remarkably higher in July than in January, which
can be explained by the sourtherly monsoon and stronger vertical mixing in summer that
favors inter-regional transport of air pollutants. We also examine the contributions of
emissions outside the BTH region to $PM_{2.5}$ concentrations in the five target regions. The
results reveal that these emissions contribute 24-33% of the 4-month mean $PM_{2.5}$
concentrations, among which more than 80% could be attributed to precursor emissions.
Among the four months, the contribution of emissions outside BTH is considerably smaller in
January (12-21%) as compared to other months (29-38%).
### 3.3  Response of $PM_{2.5}$ chemical components to emissions of air pollutants
Ambient $PM_{2.5}$ is comprised of complicated chemical components with distinctly different
formation pathways. To gain deeper insight into the formation mechanisms and source
attribution of $PM_{2.5}$, we examine the sensitivities of major $PM_{2.5}$ components, including $NO_3^-$,
$SO_4^{2-}$, and OA, to stepped control of individual air pollutants, as shown in Fig. 7 (January and
July) and Fig. S7 (March and October). $NO_3^-$ concentrations are most sensitive to $NH_3$
emissions in all months except July, when the sensitivities of $NO_3^-$ concentrations to $NH_3$ and
$NO_X$ emissions are similar. The $NO_3^-$ sensitivities to $NO_X$ emissions differ significantly
according to season. In most months, $NO_3^-$ concentrations are positively correlated with $NO_X$
emissions. In January, however, the sensitivities of $NO_3^-$ concentrations to $NO_X$ emissions are
mostly negative and could be positive at large reduction ratios, which can be explained by a
very strong NMVOC-limited photochemical regime, and abundant ice water for
heterogeneous formation of $HNO_3$ from $N_2O_5$ at cold temperatures. The sensitivites of $NO_3^-$ to
both $NH_3$ and $NO_X$ emissions show pronounced increasing trends with the increase of
reduction ratio, in agreement with the strong nonlinearity in these two pollutants described in
Section 3.2. NMVOC emissions make moderate positive contributions to $NO_3^-$, with the
largest and smallest contributions occuring in January and July in conjunction with NMVOC-
limited and $NO_X$-limited photochemical regimes, respectively. Finally, $SO_2$ emissions have
very small influences on $NO_3^-$ concentrations.
For $SO_4^{2-}$, $SO_2$ emissions represent the dominant contributor in all months. The sensitivity
of $SO_4^{2-}$ concentrations to $SO_2$ emissions does not change significantly with respect to
reduction ratio, consistent with the results shown in Section 3.2. The contributions of $NH_3$
emissions to $SO_4^{2-}$ concentrations are quite small except in October, when $NH_3$ accounts for
approximately one fourth the contribution of $SO_2$. $NO_X$ emissions affect $SO_4^{2-}$ concentrations
mainly by altering $O_3$ and $HO_X$ concentrations, the effects of which are positive in July at
large reduction ratios, and mostly negative in other months. NMVOC emissions can impose
small impact on $SO_4^{2-}$ concentrations primarily through changing $O_3$ and $HO_X$ concentrations.
The emissions of POA and NMVOC+IVOC are obviously two major contributors to OA
concentrations. The relative importance of the two is strongly dependent on season. In July,
POA and NMVOC+IVOC make similar contributions to OA concentrations, while POA
usually contributes more in other months. In January, the contribution of POA could account
for about four times those of NMVOC+IVOC. The higher relative contribution of POA
emissions in January can be explained by several reasons. First, the POA emissions are
relatively higher in January due to residential heating, while the NMVOC emissions from
solvent use and biogenic sources are higher in July. Second, lower temperature in winter
favors the partitioning of the semi-volatile components comprising POA to the particle phase,
whereas higher temperature and stronger radiation in July accelerate the formation of SOA
from NMVOC+IVOC. Similar to $SO_4^{2-}$, the impact of $NO_X$ emissions on OA concentrations
also works through two pathways. Besides the abovementioned photochemical pathway, $NO_X$
emission reductions could lead to OA increases due to the fact that SOA yield, defined as the
ratio of SOA formation to the consumption of a precursor, is generally higher at a low-$NO_X$
condition than at a high-$NO_X$ condition. As an integrated effect, the responses of OA
concentrations to $NO_X$ emissions are negative in most situations.
**3.4   $PM_{2.5}$ responses to emission reductions during heavy-pollution episodes**
Having shown the responses of monthly-mean $PM_{2.5}$ concentrations to pollutant emissions,
we are also interested in heavy-pollution episodes, in which the source contributions could be
quite different from the monthly-mean results, largely due to variations in meteorological
conditions. To provide more insight into the control strategies for heavy pollution, we use the
ERSM technique to investigate the source contribution features during three typical heavy-
pollution episodes. We first select 47 heavy-pollution episodes over the BTH region during
2013-2015 (Table S7). Subsequently, we employ the Hybrid Single Particle Lagrangian
Integrated Trajectory (HYSPLIT) model (Stein et al., 2015) and Concentration Weighted
Trajectory (CWT) method (Cheng et al., 2013) to identify the potential source regions for
$PM_{2.5}$ during each episode, and categorize these episodes according to their source regions.
We then select a representative episode from each of three most important pollution types in
which the air mass primarily originates from local areas ("Local" type), from the south
("South" type), and from the southeast ("Southeast" type). We give preference to episodes
within the four-month simulation period of this study to facilitate a comparison with the
monthly-mean source contribution features. For this reason, we select (1) January 5-7, 2014,
(2) October 7-11, 2014, and (3) October 29-31, 2014 as representatives of the "Local",
"South", and "Southeast" types. The selection of heavy-pollution episodes is detailed in
Section 2 of the Supplementary Information.

4        Figure 8 shows the contribution of precursor and primary inorganic $PM_{2.5}$ emissions from

individual regions to $PM_{2.5}$ concentrations during the three heavy-pollution episodes, and Fig.
9 illustrates the sensitivity of $PM_{2.5}$ concentrations to stepped control of individual pollutant-
sector combinations. During January 5-7, 2014 ("Local" type), the contributions of local
emission sources to $PM_{2.5}$ concentrations far exceed those from other regions within BTH as
well as from outside of BTH (Fig. 8). In contrast to the monthly mean results (Section 3.2),
the contributions of primary inorganic $PM_{2.5}$ emissions are comparable to, and even larger
than those of precursor emissions in the BTH region. The total contributions of primary $PM_{2.5}$
(including POA) account for as high as 70-80% of the contributions of all pollutants within
the BTH region, which highlights the crucial importance of primary $PM_{2.5}$ controls during this
episode. Moreover, the controls of NMVOC, $NH_3$, and $SO_2$ emissions could contribute
moderately to reducing $PM_{2.5}$ concentrations. However, $NO_X$ emission reduction induces an
increase in $PM_{2.5}$ concentrations, even at an 80% reduction ratio. Therefore, effective
temporary control measures for this episode should focus on the controls of local emissions,
with emphasis laid on primary $PM_{2.5}$.

19       During October 7-11, 2014 ("South" type), the contributions of emissions outside BTH to

$PM_{2.5}$ concentrations are as large as 33% in Beijing, and 40-50% in other regions. Within the
BTH region, the emissions from Southern Hebei can have similar effects to local emissions
on $PM_{2.5}$ concentrations in Beijing, indicating a strong long-range transport from the south. In
addition, the total contributions of precursor emissions about double those of primary
inorganic $PM_{2.5}$ emissions. Among all precursors, $PM_{2.5}$ concentrations are mainly sensitive to
emissions of $NH_3$, NMVOC+IVOC, and POA. The sensitivity of $PM_{2.5}$ concentrations to $NO_X$
emissions increases dramatically with reduction ratio. Although small $NO_X$ reductions may
slightly elevate $PM_{2.5}$ concentrations, large $NO_X$ emission reduction (> 50%) can result in
significant $PM_{2.5}$ reduction. To effectively mitigate $PM_{2.5}$ pollution during this episode, we
should implement control measures for precursor emissions in both the BTH region
(especially the southern part) and regions south of BTH. The $NO_X$ emissions, if controlled,
should be reduced by at least 50% to avoid adverse side effect.

For October 29-31, 2014 ("Southeast" type), $PM_{2.5}$ concentrations are also significantly affected by emissions outside the BTH region. Within the BTH region, the $PM_{2.5}$ concentrations in Beijing and Northern Hebei are about equally affected by local emissions and emissions from Eastern Hebei and Southern Hebei, while local emissions play dominant roles in other regions. The emissions of both precursor and primary inorganic $PM_{2.5}$ within the BTH region make important contributions to $PM_{2.5}$ concentrations, and the relative significance of the two is dependent on region. All precursors except $NO_X$ can contribute considerably to $PM_{2.5}$ reductions, and the sensitivity of $PM_{2.5}$ to $NH_3$ increase rapidly with emission ratio. $NO_X$ emissions are negatively correlated with $PM_{2.5}$ concentrations in most cases. Regarding the temporary control strategy for this episode, it is preferable to implement joint controls of primary $PM_{2.5}$ and precursors both within and outside the BTH region, with stringent measures over the Eastern and Southern Hebei.

From the analysis above, we conclude that the source contributions are tremendously different in these three episodes, which have been demonstrated to represent some key features of the corresponding pollution types ("Local", "South", and "Southeast" types). Therefore, episode-specific control strategies need to be formulated based on the source contribution features of individual pollution types. Nevertheless, the results of this study are not yet sufficient to guide the development of temporary control strategies for all heavy-pollution episodes, because the conclusions drawn from the three episodes may not be generalized to pollution types. In future studies, we need to simulate more episodes to improve their classification and to comprehensively understand the source contribution features of each pollution type. For a coming heavy-pollution episode, we can predict its pollution type using an air quality forecasting model, and subsequently formulate the temporary control strategies based on the source contribution features of this specific pollution type.

## 4 Conclusion and implications

In the present study, we investigated the nonlinear response of $PM_{2.5}$ concentrations to emission changes of multiple pollutants from different sectors and regions over the BTH region, using the ERSM technique coupled with the CMAQ/2D-VBS model.

Among all pollutants, primary inorganic $PM_{2.5}$ makes the largest contribution (24-36%) to the 4-month mean $PM_{2.5}$ concentrations. The contribution from primary inorganic $PM_{2.5}$ is especially high in heavily polluted winter, and is dominated by the industry as well as

residential and commercial sectors. The total contributions of all precursors to $PM_{2.5}$ concentrations range between 31% and 48%. Among the precursors, $PM_{2.5}$ concentrations are primarily sensitive to the emissions of $NH_3$, NMVOC+IVOC, and POA. With the increase of reduction ratio, the sensitivities of $PM_{2.5}$ concentrations to pollutant emissions remain roughly constant for primary inorganic $PM_{2.5}$ and $SO_2$, increase substantially for $NH_3$ and $NO_X$, and decrease slightly for POA and NMVOC+IVOC. The contributions of primary inorganic $PM_{2.5}$ emissions to $PM_{2.5}$ concentrations are dominated by local emission sources, which account for over 75% of the total primary inorganic $PM_{2.5}$ contributions. For precursors, however, emissions from other regions could play similar roles to local emission sources in the summer and over the northern part of BTH. Different $PM_{2.5}$ chemical components are associated with distinct source contribution features. The $NO_3^-$ and $SO_4^{2-}$ concentrations are most sensitive to emissions of $NH_3$ and $SO_2$, respectively. The emissions of the POA and NMVOC+IVOC are two major contributors to OA concentrations, with their relative importance depending on season.

The source contribution features are significantly different for three typical heavy-pollution episodes, which belong to three distinct pollution types. The $PM_{2.5}$ concentrations in the first episode ("Local" type) are dominated by local sources and primary $PM_{2.5}$ emissions, while the second episode ("South" type) is primarily affected by precursor emissions from local and southern regions. The third episode ("Southeast" type) is significantly influenced by emissions of both primary inorganic $PM_{2.5}$ and precursors from multiple regions. Future investigations are needed to acquire generalized patterns for the source contributions of various heavy-pollution types.

The results of the present study have important implications for $PM_{2.5}$ control policies over the BTH region. First, the controls of primary $PM_{2.5}$ emissions should be a priority in $PM_{2.5}$ control strategies. Primary $PM_{2.5}$, including primary inorganic $PM_{2.5}$ and POA, contribute over half of the 4-month mean $PM_{2.5}$ concentrations, which is even higher in the winter when heavy pollution frequently occurs. The industry sector and the residential and commercial sectors represent 85% of the total primariy $PM_{2.5}$ emissions, and therefore should be the focus of primary $PM_{2.5}$ controls. In particular, we should pay special attention to the residential and commercial sectors, which account for half of the total contribution of primary $PM_{2.5}$ emissions to $PM_{2.5}$ concentrations in the winter but have been frequently neglected in China's previous control policies. Second, the control policies for NMVOC and IVOC

emissions should be strengthened. The sensitivity of $PM_{2.5}$ concentrations to NMVOC+IVOC is one of the largest among all precursors. In particular, the controls of NMVOC and IVOC emissions are very effective for $PM_{2.5}$ reduction even at the initial control stage, as indicated by the large sensitivity at small reduction ratios. Moreover, NMVOC reduction is also crucial for the mitigation of $O_3$ pollution considering a NMVOC-limited regime over the urban and its surrounding areas (Xing et al., 2011). Third, $NO_X$ emissions should be substantially reduced in both the BTH and other parts of China; in the long run, the reduction ratio should preferably approach their maximum feasible reduction levels. Fourth, more stringent control policies should be enforced in Southern Hebei, which on one hand suffers from the most severe $PM_{2.5}$ pollution (Wang et al., 2014b), and on the other hand, significantly affects both local and regional $PM_{2.5}$ concentrations. Last but not least, considering the distinct source contributions in different heavy pollution episodes, episode-specific temporary control strategies should be formulated according to the source contribution feature of the specific pollution type.

The present study has a few limitations. First, the establishment of ERSM requires several hundred or over 1000 emission scenarios, although the scenario number needed for a specific number of control variables has already been dramatically reduced as compared to the conventional RSM technique. Studies are needed to further reduce the scenario number but retain the accuracy of the ERSM technique. Second, the current ERSM is developed based on the meteorological conditions simulated for the base year, and has not considered the impact of inter-annual variations in meteorological conditions on the relationships between emissions and $PM_{2.5}$ concentrations. Third, although the ERSM-predicted responses of $PM_{2.5}$ concentrations to precursor emissions have been demonstrated to agree well with chemical transport model simulations, evaluating the predicted responses against the actual situation in the real atmosphere still represents a major challenge, because it is extremely difficult to artificially perturb emissions in the atmosphere. Last but not the least, the NMVOC and IVOC emissions have been lumped together in this study to reduce the number of control variables. Considering their differences in sources and SOA formation potentials (Jathar et al., 2014; Wu et al., 2017), a detailed quantification of the individual contributions of NMVOC and IVOC emissions from various sources to $PM_{2.5}$ concentrations is required in the future to better inform NMVOC/IVOC control policies.

## Acknowledgements

This research has been supported by National Science Foundation of China (21625701 & 21521064), MOST National Key R & D program (2016YFC0207601), Strategic Pilot Project of Chinese Academy of Sciences (XDB05030401), the UCLA Sustainable Los Angeles Grand Challenge 2016 YZ-50958, and the Jet Propulsion Laboratory, California Institute of Technology, under contract with NASA. The simulations were completed on the "Explorer 100" cluster system of Tsinghua National Laboratory for Information Science and Technology.

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

**Tables and figures**
Table 1. Description of the RSM/ERSM prediction systems developed in this study.

| Method | Control variables | Control scenarios |
|---|---|---|
| Conventional RSM technique | 5 control variables:<br>total emissions of $NO_X$, $SO_2$, $NH_3$, NMVOC+IVOC, and POA | 101 control scenarios:<br>1) 1 CMAQ/2D-VBS base case;<br>2) 100[a] scenarios generated by applying LHS method for the 5 variables. |
| ERSM technique | 55 control variables in total:<br>11 control variables in each of the 5 regions, including 7 nonlinear control variables, i.e.,<br>1) $NO_X$/large point sources (LPS)[b]<br>2) $NO_X$/other sources<br>3) $SO_2$/LPS<br>4) $SO_2$/other sources<br>5) $NH_3$/all sources<br>6) NMVOC+IVOC/all sources<br>7) POA/all sources<br>and 4 linear control variables, i.e.,<br>8) Primary inorganic $PM_{2.5}$/power plants<br>9) Primary inorganic $PM_{2.5}$/Industry<br>10) Primary inorganic $PM_{2.5}$/residential & commercial<br>11) Primary inorganic $PM_{2.5}$/transportation | 1121 control scenarios:<br>1) 1 CMAQ/2D-VBS base case;<br>2) 1000 scenarios, including 200[a] scenarios generated by applying LHS method for the nonlinear control variables in Beijing, 200 scenarios generated in the same way for Tianjin, 200 scenarios for Northern Hebei, 200 scenarios for Southern Hebei, and 200 scenarios for Eastern Hebei;<br>3) 100[a] scenarios generated by applying LHS method for the total emissions of $NO_X$, $SO_2$, $NH_3$, NMVOC+IVOC, and POA;<br>4) 20 scenarios where one primary inorganic $PM_{2.5}$ control variable is set to 0.25 for each scenario. |

[a] 100 and 200 scenarios are needed for the response surfaces for 5 and 7 variables, respectively (Xing et al.,
2011; Wang et al., 2011).
[b] LPS includes power plants, iron and steel plants, and cement plants

Table 2. Comparison between ERSM-predicted and CMAQ/2D-VBS-simulated $PM_{2.5}$ concentrations for 54 out-of-sample scenarios.

| Month | Variable | Statistical index | Beijing | Tianjin | Northern Hebei | Eastern Hebei | Southern Hebei |
|---|---|---|---|---|---|---|---|
| Jan | $PM_{2.5}$ concentration | R | 0.998 | 0.998 | 0.995 | 0.997 | 0.997 |
| | | MNE (%) | 0.52 | 0.55 | 0.64 | 0.67 | 0.60 |
| | | Maximum NE (%) | 7.56 | 6.98 | 10.67 | 8.01 | 8.03 |
| | | 95% percentile of NEs (%) | 1.61 | 2.86 | 2.92 | 3.46 | 3.02 |
| | | NME (%) | 0.44 | 0.46 | 0.57 | 0.53 | 0.53 |
| | $PM_{2.5}$ response | R | 0.998 | 0.998 | 0.995 | 0.997 | 0.997 |
| | | NME (%) | 3.36 | 3.48 | 4.25 | 4.00 | 3.88 |
| Mar | $PM_{2.5}$ concentration | R | 0.999 | 0.996 | 0.998 | 0.995 | 0.999 |
| | | MNE (%) | 0.37 | 0.54 | 0.39 | 0.57 | 0.49 |
| | | Maximum NE (%) | 3.75 | 6.58 | 4.30 | 5.04 | 3.22 |
| | | 95% percentile of NEs (%) | 1.53 | 3.15 | 2.03 | 4.35 | 2.03 |
| | | NME (%) | 0.31 | 0.45 | 0.34 | 0.49 | 0.42 |
| | $PM_{2.5}$ response | R | 0.999 | 0.996 | 0.998 | 0.995 | 0.999 |
| | | NME (%) | 2.38 | 4.32 | 2.70 | 4.55 | 3.59 |
| Jul | $PM_{2.5}$ concentration | R | 0.997 | 0.998 | 0.998 | 0.999 | 0.999 |
| | | MNE (%) | 0.94 | 0.54 | 0.46 | 0.37 | 0.47 |
| | | Maximum NE (%) | 5.05 | 5.02 | 4.65 | 1.83 | 3.62 |
| | | 95% percentile of NEs (%) | 3.47 | 2.33 | 2.17 | 1.49 | 1.87 |
| | | NME (%) | 0.80 | 0.47 | 0.41 | 0.33 | 0.39 |
| | $PM_{2.5}$ response | R | 0.997 | 0.998 | 0.998 | 0.999 | 0.999 |
| | | NME (%) | 4.97 | 3.71 | 2.80 | 2.58 | 2.78 |
| Oct | $PM_{2.5}$ concentration | R | 0.996 | 0.994 | 0.999 | 0.999 | 0.999 |
| | | MNE (%) | 0.83 | 0.70 | 0.36 | 0.39 | 0.36 |
| | | Maximum NE (%) | 8.90 | 11.19 | 3.79 | 3.90 | 2.46 |
| | | 95% percentile of NEs (%) | 3.04 | 3.50 | 1.44 | 2.10 | 1.64 |
| | | NME (%) | 0.67 | 0.58 | 0.30 | 0.35 | 0.32 |
| | $PM_{2.5}$ response | R | 0.996 | 0.994 | 0.999 | 0.999 | 0.999 |
| | | NME (%) | 4.51 | 5.64 | 2.20 | 3.29 | 2.79 |

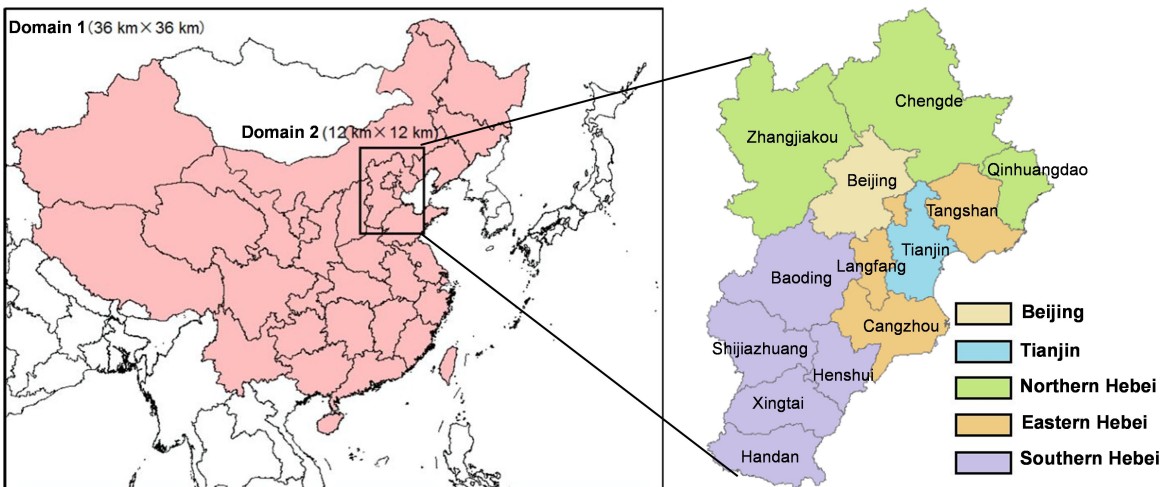

Figure 1. Double nesting domains used in CMAQ/2D-VBS simulation (left) and the definition of five target regions in the innermost domain, denoted by different colours (right). The grey lines in the right figure represent the boundaries of prefecture-level cities.

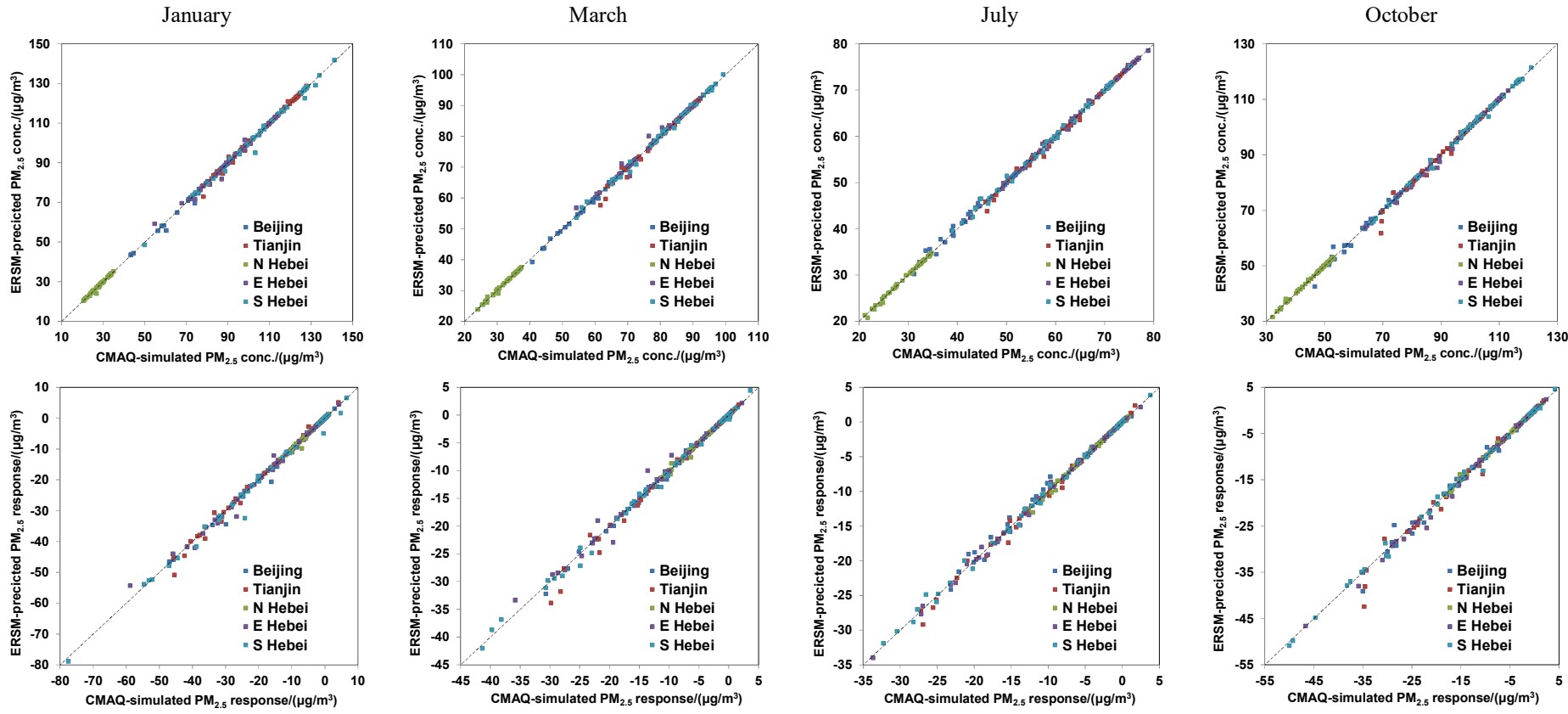

1    Figure 2. Comparison of PM$_{2.5}$ concentrations (top row) and PM$_{2.5}$ responses (bottom row) predicted by the ERSM technique with out-of-

2    sample CMAQ/2D-VBS simulations. The dashed line is the one-to-one line indicating perfect agreement.

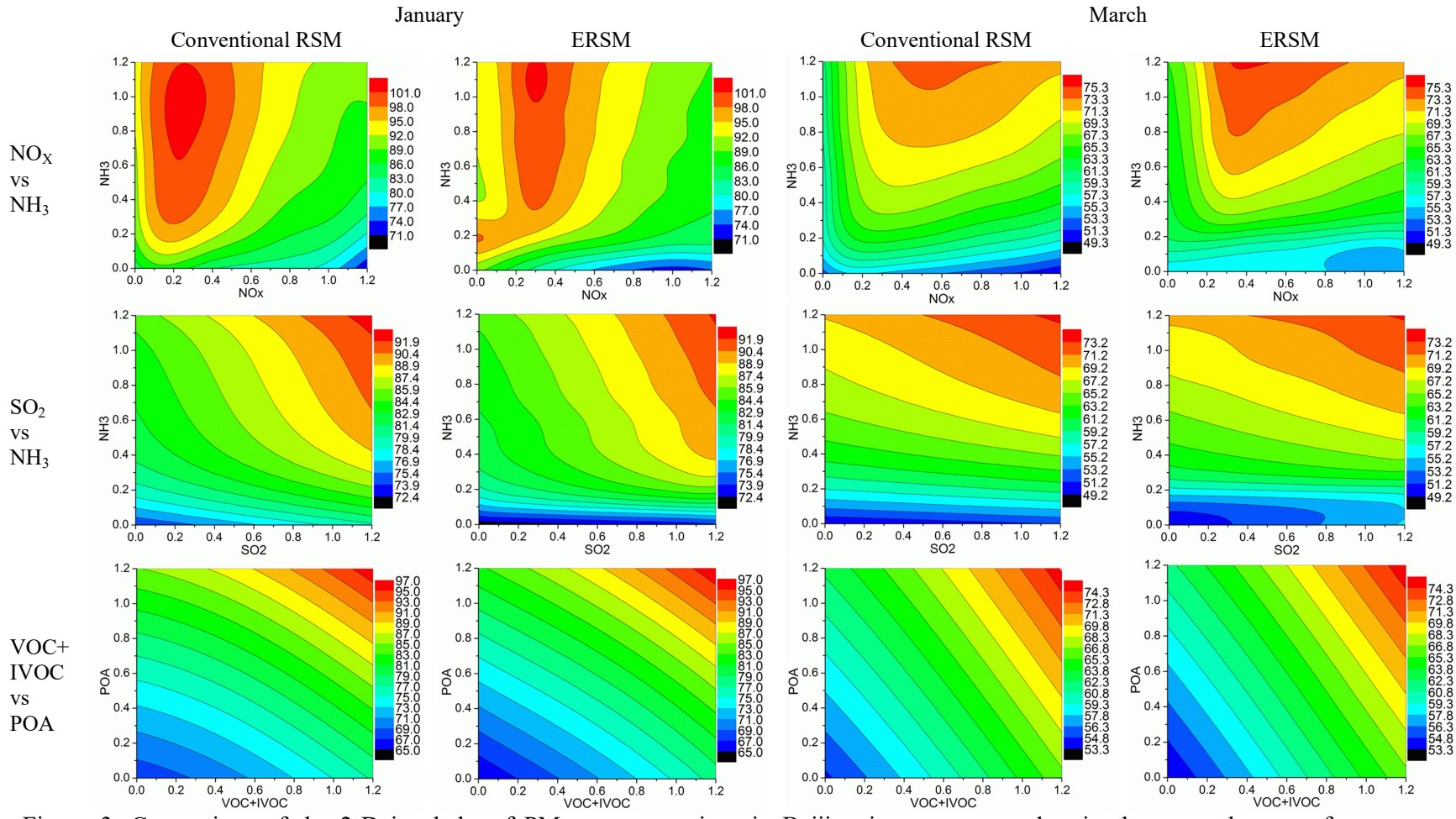

Figure 3. Comparison of the 2-D isopleths of $PM_{2.5}$ concentrations in Beijing in response to the simultaneous changes of precursor emissions in all five regions derived from the conventional RSM technique and the ERSM technique. The X- and Y-axis represent the emission ratio, defined as the ratios of the changed emissions to the emissions in the base case. The colour contours represent $PM_{2.5}$ concentrations (unit: $\mu g\ m^{-3}$).

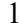

Figure 3. Continued.

Figure 3. Continued.

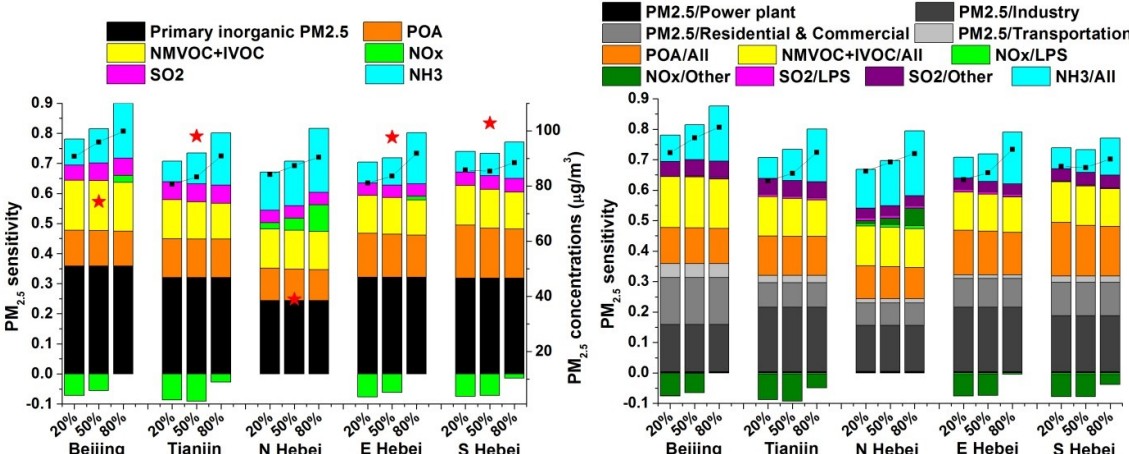

Figure 4. Sensitivity of 4-month mean PM$_{2.5}$ concentrations to stepped control of individual air pollutants (left) and individual pollutant-sector combinations (right). The X-axis shows the reduction ratio (= 1 – emission ratio). The Y-axis shows PM$_{2.5}$ sensitivity, which is defined as the change ratio of concentration divided by the reduction ratio of emissions. The coloured bars denote the PM$_{2.5}$ sensitivities when a particular emission source is controlled while the others stay the same as the base case; the black dotted line denotes the PM$_{2.5}$ sensitivity when all emission sources are controlled simultaneously. The red stars represent PM$_{2.5}$ concentrations in the base case.

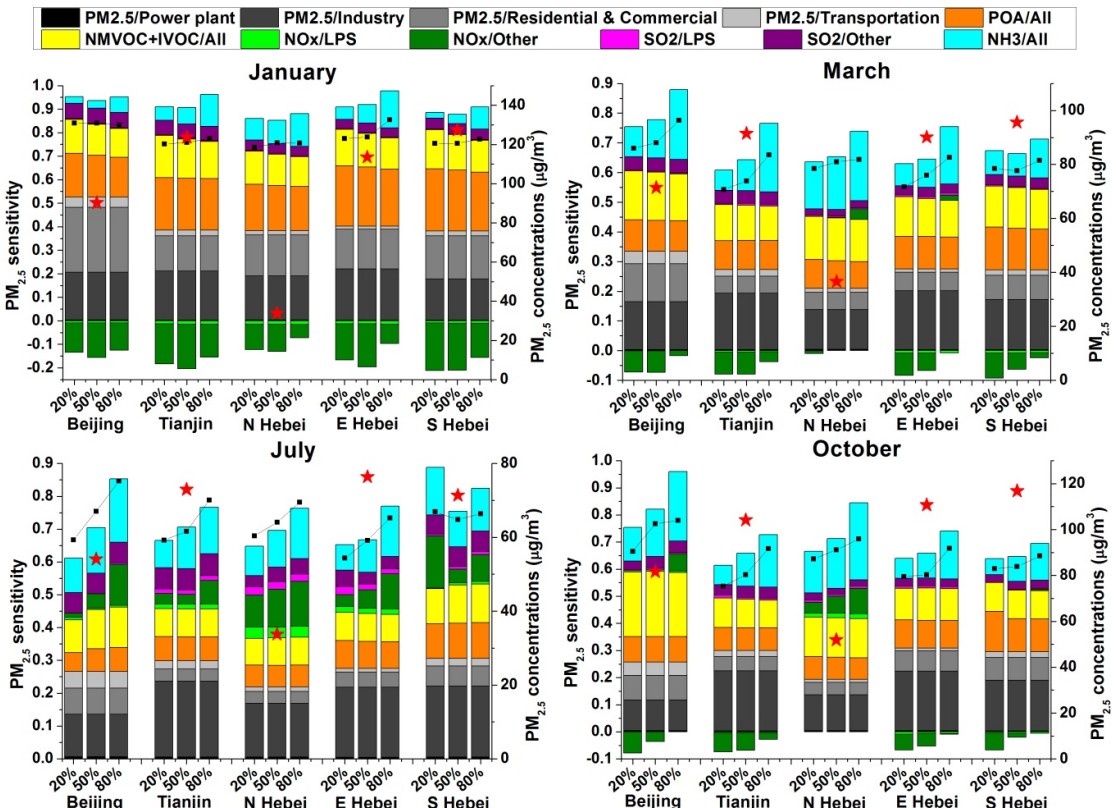

Figure 5. Sensitivity of monthly mean PM$_{2.5}$ concentrations to stepped control of individual air pollutants from individual sectors in January, March, July, and October. The meanings of X-axis, Y-axis, coloured bars, black dotted lines, and red stars are the same as Fig. 4.

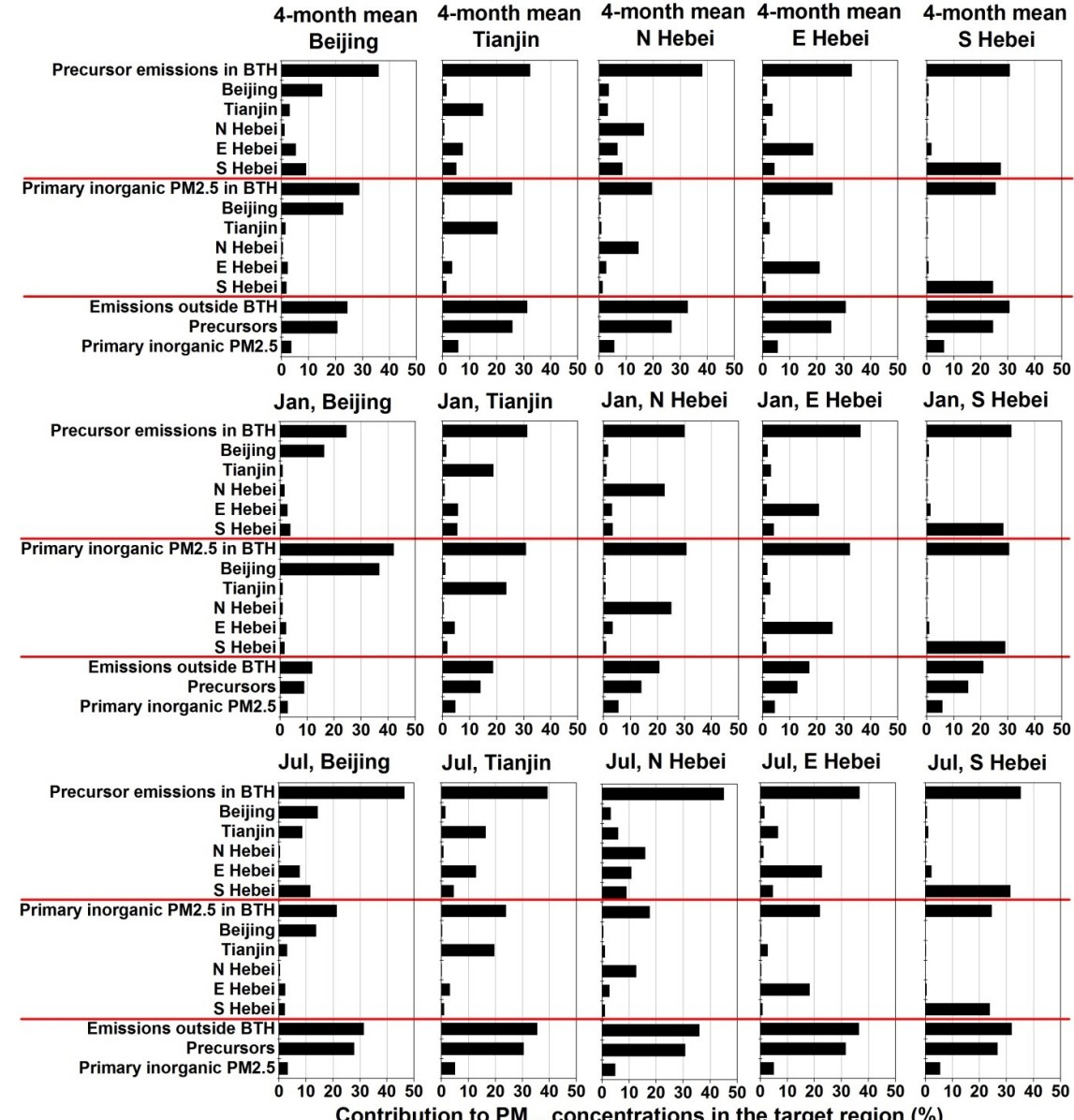

Figure 6. Contributions of precursor (NO$_X$, SO$_2$, NH$_3$, NMVOC, IVOC, and POA) and
primary inorganic PM$_{2.5}$ emissions from individual regions to PM$_{2.5}$ concentrations. The
contributions are quantified by comparing the base case with sensitivity scenarios in which
emissions from a specific source are reduced by 80%. This figure illustrates contributions to
4-month mean PM$_{2.5}$ concentrations and monthly mean PM$_{2.5}$ concentrations in January and
July. The results for March and October are given in Fig. S6.

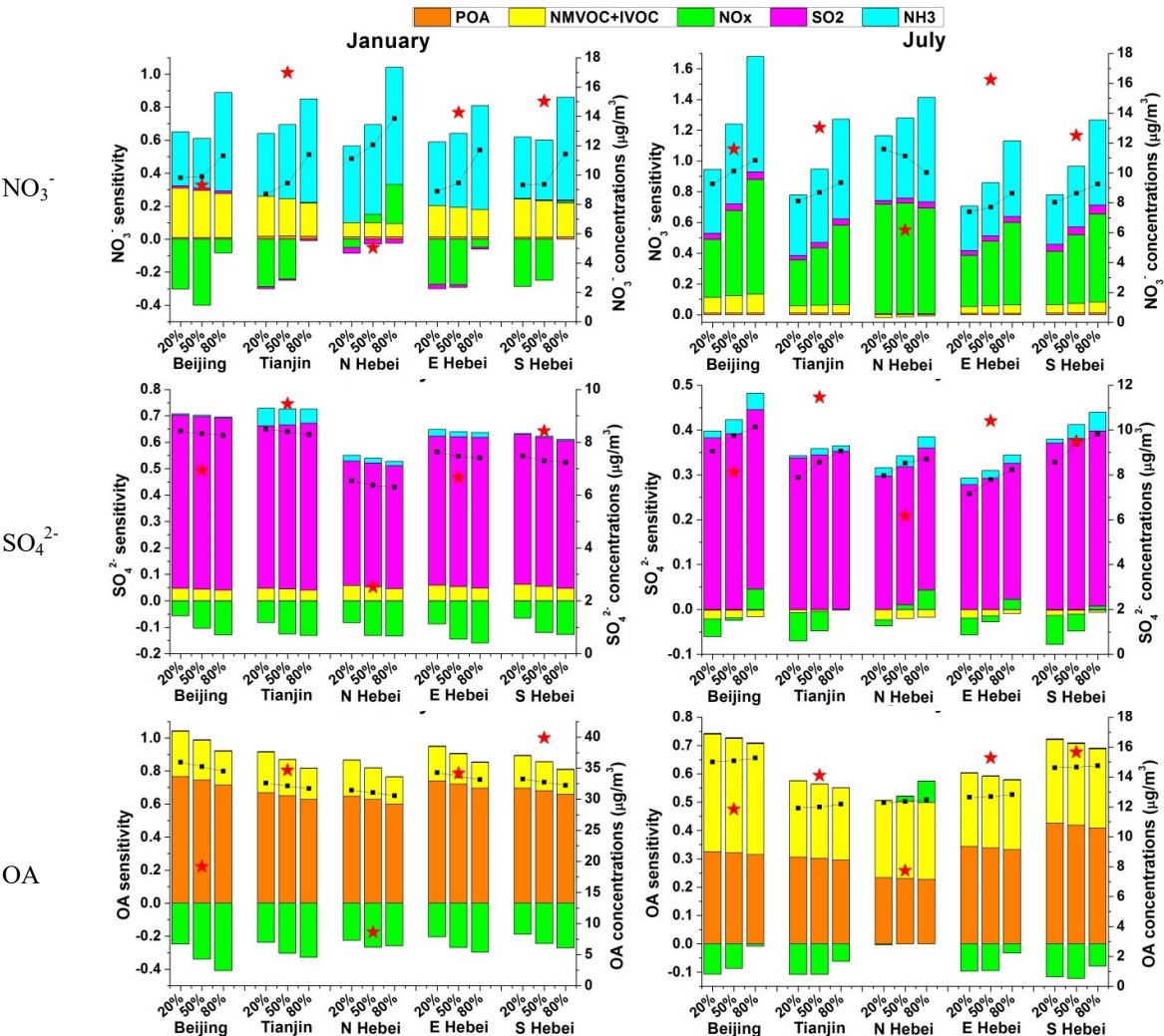

Figure 7. Sensitivity of monthly mean $NO_3^-$, $SO_4^{2-}$, and OA concentrations to stepped control of individual air pollutants in January and July. The meanings of X-axis, Y-axis, coloured bars, black dotted lines, and red stars are the same as Fig. 4 but for $NO_3^-/SO_4^{2-}$/OA. The results for March and October are given in Fig. S7.

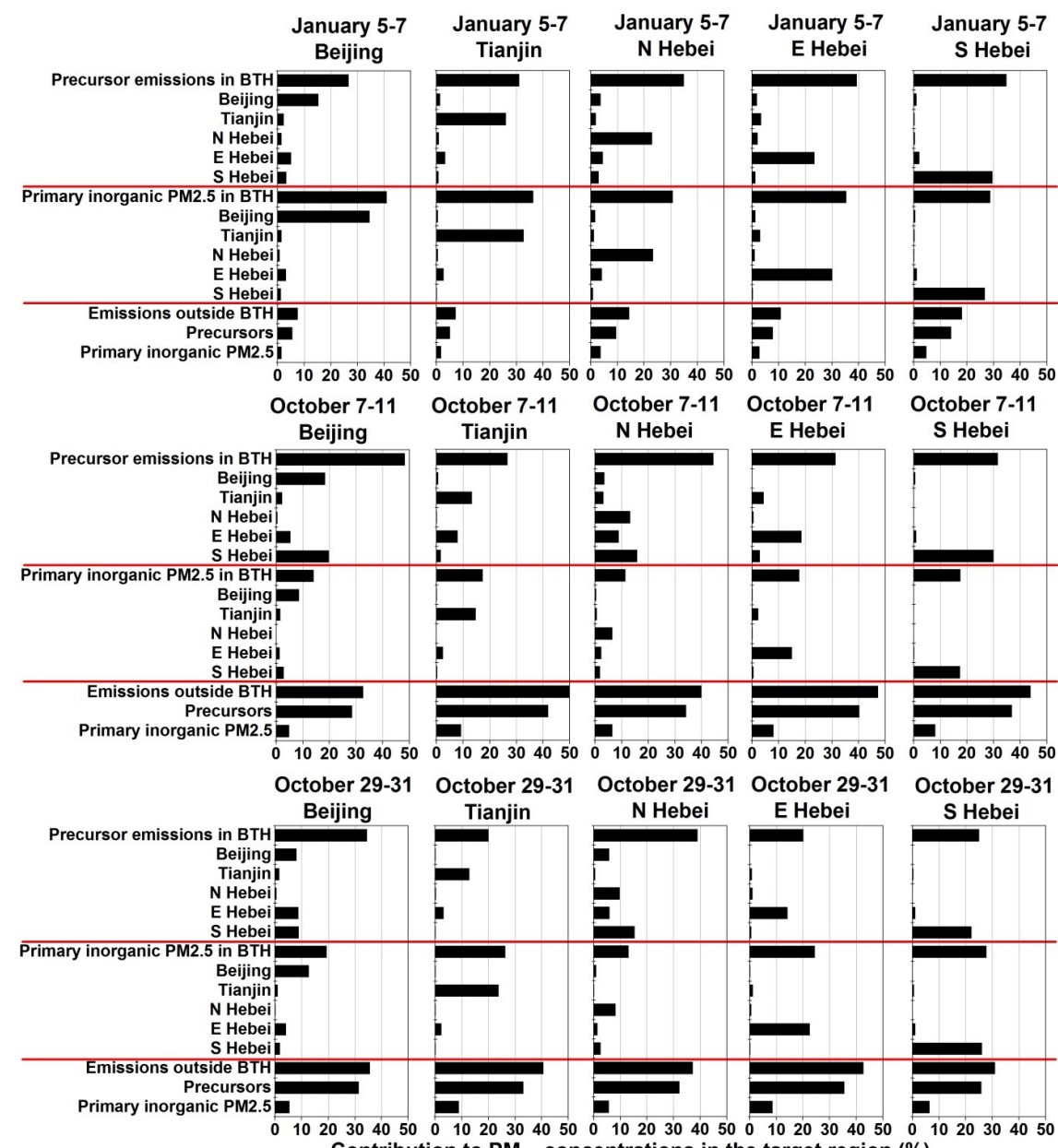

3   Figure 8. Contribution of precursor (NO$_X$, SO$_2$, NH$_3$, NMVOC, IVOC, and POA) and primary

4   inorganic PM$_{2.5}$ emissions from individual regions to PM$_{2.5}$ concentrations during three

5   typical heavy-pollution episodes.

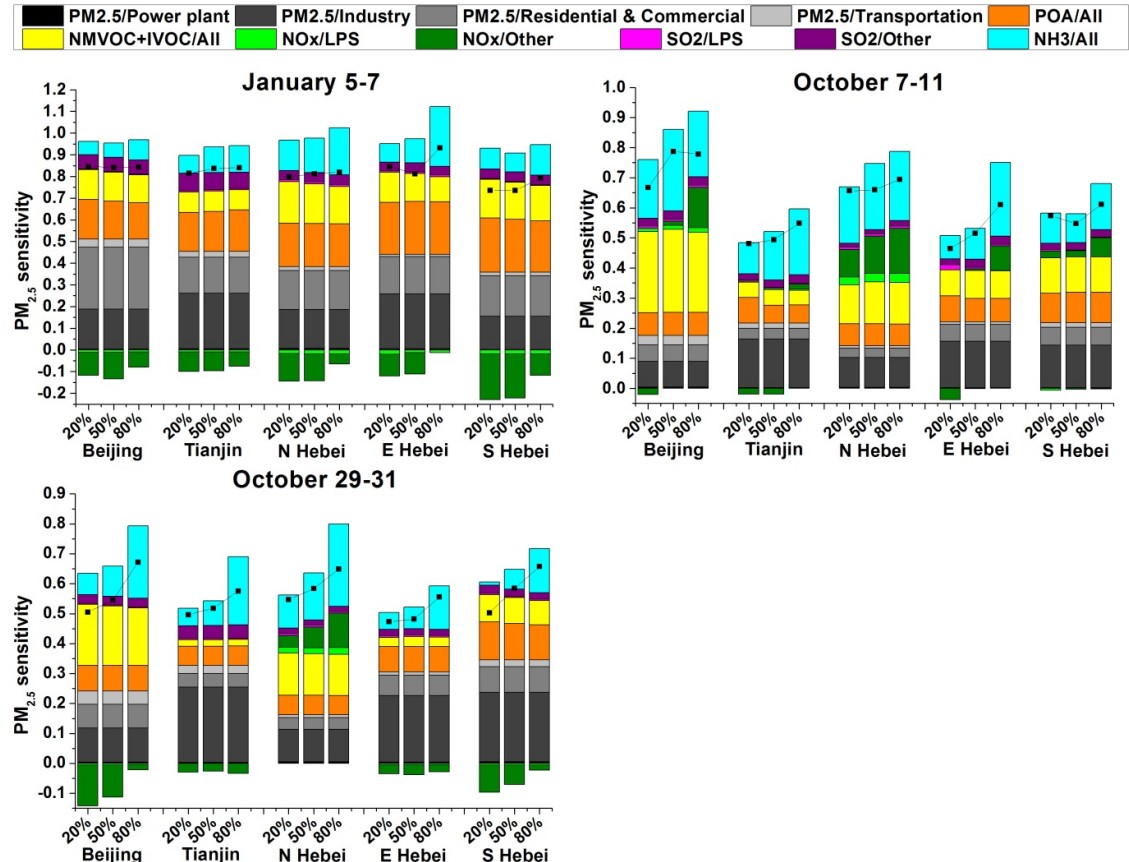

2 Figure 9. Sensitivity of PM$_{2.5}$ concentrations to stepped control of individual air pollutants
3 from individual sectors during three heavy-pollution episodes. The meanings of X-axis, Y-
4 axis, coloured bars, and black dotted lines are the same as Fig. 4.
