# Peer review of "A modeling study of the nonlinear response of fine"

_Atmospheric Chemistry and Physics, 2017_

## Referee Comment (RC1) · Anonymous Referee #3 · 26 Jun 2017

This paper used an Extended Response Surface Modeling (ERSM) technique to assess the source contributions of various chemical precursors, emission sectors, source regions, and their combinations to the PM2.5 concentrations over the BTH area. It extended the previous conventional RSM model and pursued more than 1000 simulation scenarios. It is informative and valuable to the air pollution controls over the heavily polluted BTH area. I would suggest this paper to be published after minor revision. (1) In the abstract, page 2, line 6, "primary inorganic PM2.5 is the single pollutant which makes the largest contribution (24-36%) to PM2.5 concentrations." What is the exact mean of the word "single"? (2) In the Table S4, "Statistical results for the comparison of monthly PM2.5 concentrations", the variable calculated in the statistics is hourly PM2.5

concentrations, right? (3) In Table S4 and S5, please add the number of data pairs, especially in S5. (4) I would suggest the authors add a discussion on the limitations or uncertainties of this study at the end of the conclusion section.
* * *

---

## Referee Comment (RC2) · Anonymous Referee #2 · 21 Jul 2017

General comments

ERSM has been developed by extending the capabilities of the conventional RSM. Its performance was evaluated. Then, sensitivities of emissions of various primary pollutants and precursors, sectors, and regions on seasonal concentrations of PM2.5 and their components in BTH region were discussed.

The advantage of the ERSM technique is that it can represent complex non-linear relationships between ambient pollutant concentrations and their precursor emissions. On the other hand, it requires over 1000 simulations. If changes in ambient concentrations in several future scenarios, only several simulations with the brute force method

are required. I feel the advantage of the ERSM which overcome tremendous efforts to run simulations over 1000 times is not fully emphasized in this manuscript. In addition, descriptions of limitations of the ERSM technique are scarce. Please add more descriptions on the advantage and disadvantage of the ERSM technique.

ERSM could provide valuable information to develop effective strategies based on complex non-linear relationships. It means non-linear responses should represent the actual situation in the real atmosphere. I think validation of the responses obtained by ERSM is not enough whereas comparisons of observed concentrations have been made.

Especially, nonlinear responses of NOX emissions are critical for policy making. How much NOX reduction is necessary to realize positive effects to reduce PM2.5 concentrations? ERSM could give the answer. However, if the answer is not correct in the real atmosphere, policies may fail to realize PM2.5 reductions.

Which components are included in inorganic PM2.5? Is EC included? How about other components like metals? It looks strange that primary organic aerosol (POA) is included as a precursor probably due to treatment in VBS. Please give precise definitions of these words.

What do "discrepant temporary control strategies" mean? How are they possible? I understand major sources are different in each heavy air pollution episode. However, it could be possible to implement different temporary control strategies for each episode only if it could be forecasted. Can ERSM be used to forecast major sources in coming heavy air pollution episodes? I think differences of major sources in each episode suggest to implement strategies which control emissions of all the sources which could be major in various episodes.

Specific comments

Page 3, Line 8 How much are 2012 levels?

Page 3, Line 22 The sentence here says "CTMs are the only feasible tools for evaluating the response of PM2.5 concentrations to emission changes". However, the sentences around the line 14 describe that embedding chemical tracers in chemical transport models (CTMs) cannot represent non-linear response. They may confuse some readers who are not familiar to CTMs.

Page 3, Line 26 "Sensitivities" are more appropriate than "contributions" in the context here.

Page 4, Line 5 How inadequate?

Page 5, Line 7 How were emissions of IVOC provided?

Page 5, Line 8 OA and SOA are listed parallelly, but SOA is included in OA.

Page 5, Line 30 I think NCEP final analysis data is not reanalysis data. Is it not used for grid nudging?

Page 6, Line 4 I think terrain data is not from MODIS.

Page 6, Line 25 How about open biomass burning emissions?

Page 9, Line 21 How about the performance of SO42-, NO3-, and OA?

Page 10, Line 8 Why are only NMEs shown? How about R and MNEs? I suppose it is more important for RSM to see responses than to reproduce concentrations.

Page 10, Line 15 I do not understand meaning of comparisons between ERSM and conventional RSM. Why these two model could produce different results? Which should be correct? The sentence in the line 31 says that the ERSM predictions are definitely subject to numerical errors, but I do not know why "definitely". Although there are descriptions of ERSM in the first paragraph of the section 2.2, the advantages and disadvantages of ERSM against conventional RSM should be clearly explained.

Page 11, Line 5 What is the advantage of ERSM against conventional RSM in the

results shown in Figure 4? I think the sector-wise results shown in the right figure cannot be obtained by conventional RSM. Is that correct? Please described what is newly obtained by using ERSM.

Page 11, Line 16 It looks strange to represent primary inorganic PM2.5 as "single pollutant" because it is a mixture of various components in fact.

Page 11, Line 29 What is the reasons of small sensitivities of SO2 emissions on PM2.5?

Page 11, Line 31 Nonlinear sensitivities of NOX emissions and their changes from negative to positive are described from here. I also agree that this is very important phenomena to consider effective emission controls. However, on the other hand, the descriptions in the page 10 treat such a nonlinear change in sensitivities and differences with conventional RSM as just a rare case involving large unrealistic reduction of NOx emissions. I do not agree that. Even if large NOx reduction is required, the performance of ERSM to represent such a nonlinear change should be carefully evaluated.

Page 12, Line 2 Indeed, the regimes are very important for negative and positive sensitivities of NOX emissions. Therefore, it is quite important to see if ERSM could accurately represent regimes in the real atmosphere. I suppose such validations are scarce.

Page 12, Line 20 Are there any discussions on differences between sensitivities of all pollutants and sectors and sum of sensitivities of individual pollutants and sectors?

Page 12, Line 31 What is a reason of higher sensitivities of residential and commercial sources in winter? Heating?

Page 13, Line 8 Are there any specific results indicating the importance of NOx emissions outside the BTH region?

Page 14, Line 6 How does seasonal variations of NH3 emissions look like?

Page 14, Line 25 Is it confirmed that NOX competes with SO2 for NH3 in a thermo-dynamic pathway? I think SO42- is much more predominantly in aerosol phase than NO3-.

Page 15, Line 1 Does this POA include semivolatile components which could conden-sate only under lower temperature in winter?

Page 17, Line 10 I agree more model simulations of more episodes are necessary, but a model can always give results. I believe what is important is to confirm model results are consistent with actual situations in the real atmosphere. That is quite important to consider effective strategies for heavy air pollutions.

Page 18, Line 18 I am wondering if NMVOC and IVOC should be discussed together to implement any strategies because their sources and their effects on PM2.5 and ozone could be different.

Page 18, Line 24 I agree NOx reduction is necessary in the long run. However, it could increase PM2.5 emissions in the near term with slight reduction. How should such adverse effects be considered? Any messages on this issue?

Page 18, Line 26 I feel the importance of Southern Hebei is not so discussed in the main text.

Technical corrections

Page 6, Line 17 origininally -> originally

──────────────────────────────

---

## Author Comment (AC1) · 12 Aug 2017

We thank the reviewer for the valuable comments. We have addressed all the comments and revised the manuscript accordingly. Please find the point-to-point response and revised manuscript in the attachment.

Please also note the supplement to this comment: https://www.atmos-chem-phys-discuss.net/acp-2017-428/acp-2017-428-AC1-supplement.zip

2017.

---

## Author Response (AR1)

Reviewer 3:

This paper used an Extended Response Surface Modeling (ERSM) technique to assess the
source contributions of various chemical precursors, emission sectors, source regions, and
their combinations to the PM$_{2.5}$ concentrations over the BTH area. It extended the previous
conventional RSM model and pursued more than 1000 simulation scenarios. It is informative
and valuable to the air pollution controls over the heavily polluted BTH area. I would suggest
this paper to be published after minor revision.

Response: We appreciate the reviewer's valuable comments which help us improve the
quality of the manuscript. We have carefully revised the manuscript according to the
reviewers' comments. Point-to-point responses are given below. The original comments are in
black, while our responses are in blue.

(1) In the abstract, page 2, line 6, "primary inorganic PM$_{2.5}$ is the single pollutant which
makes the largest contribution (24-36%) to PM$_{2.5}$ concentrations." What is the exact mean of
the word "single"?

Response: We thank the reviewer for this valuable comment. In the context of this sentence, a
"single" pollutant means one of the six pollutants (or pollutant groups) considered in the
RSM/ERSM prediction systems, i.e., NO$_X$, SO$_2$, NH$_3$, NMVOC+IVOC, POA, and primary
inorganic PM$_{2.5}$. Primary inorganic PM$_{2.5}$ is defined as the chemical components of primary
PM$_{2.5}$ other than POA. To avoid confusion, we have revised the preceding sentence as follows
in the revised manuscript (Page 2, Line 5-7).

Among all air pollutants, primary inorganic PM$_{2.5}$ makes the largest contribution (24-36%)
to PM$_{2.5}$ concentrations.

(2) In the Table S4, "Statistical results for the comparison of monthly PM$_{2.5}$ concentrations",
the variable calculated in the statistics is hourly PM$_{2.5}$ concentrations, right?

Response: The original data used in the statistical analysis are daily PM$_{2.5}$ concentrations. We
have clarified this in the footnote of the revised table.

(3) In Table S4 and S5, please add the number of data pairs, especially in S5.

Response: We have added the number of data pairs used in statistics in the revised Table S4
and S5.

(4) I would suggest the authors add a discussion on the limitations or uncertainties of this study at the end of the conclusion section.

Response: Following the reviewer's suggestion, we have added a paragraph about the limitations and uncertainties of the present study at the end of the manuscript (Page 21, Line 13-27). The added paragraph is shown as follows.

The present study has a few limitations. First, the establishment of ERSM requires several hundred or over 1000 emission scenarios, although the scenario number needed for a specific number of control variables has already been dramatically reduced as compared to the conventional RSM technique. Studies are needed to further reduce the scenario number but retain the accuracy of the ERSM technique. Second, the current ERSM technique has not considered the impact of meteorological variations on ambient concentrations. Third, although the responses of $PM_{2.5}$ concentrations to precursor emissions predicted by ERSM have been demonstrated to agree well with chemical transport model simulations, evaluating the predicted responses against the actual situation in the real atmosphere still represents a major challenge, because it is extremely difficult to artificially perturb emissions in the atmosphere. Last but not the least, the NMVOC and IVOC emissions have been lumped together in this study to reduce the number of control variables. Considering their differences in sources and SOA formation potentials, a detailed quantification of the individual contributions of NMVOC and IVOC emissions from various sources to $PM_{2.5}$ concentrations is required in the future to better inform NMVOC/IVOC control policies.

Reviewer 2:

We thank the referee for a thoughtful and detailed review of our manuscript. Incorporation of
the reviewer's suggestions has led to a much improved manuscript. Below we provide a
point-by-point response to the reviewer's comments and summarize the changes that have
been incorporated in the revised manuscript.

General comments

ERSM has been developed by extending the capabilities of the conventional RSM. Its
performance was evaluated. Then, sensitivies of emissions of various primary pollutants and
precursors, sectors, and regions on seasonal concentrations of PM2.5 and their components in
BTH region were discussed.

The advantage of the ERSM technique is that it can represent complex non-linear
relationships between ambient pollutant concentrations and their precursor emissions. On the
other hand, it requires over 1000 simulations. If changes in ambient concentrations in several
future scenarios, only several simulations with the brute force method are required. I feel the
advantage of the ERSM which overcome tremendous efforts to run simulations over 1000
times is not fully emphasized in this manuscript. In addition, descriptions of limitations of the
ERSM technique are scarce. Please add more descriptions on the advantage and disadvantage
of the ERSM technique.

Response: We appreciate the reviewer's valuable comment. The ERSM technique has several
advantages over the traditional brute force method. First, the ERSM technique is able to
characterize the nonlinearity in the relationships between ambient concentrations and air
pollutant emissions. Second, cost-effective emission controls need to optimize over various
pollutants from multiple regions and sectors. Using the brute force method, we need to
repeatedly adjust the control option combinations and run the chemical transport model for
numerous times. In contrast, the ERSM prediction system, once built, enables real-time
prediction of $PM_{2.5}$ concentrations for any given control strategy and proves to be an efficient
and user-friendly decision making tool. Third, ERSM can be applied to design least-cost
control strategy once it is coupled with control cost models/functions that links the emission
reductions with economic costs.

The major disadvantage of the ERSM technique is that it requires several hundred or over
1000 emission scenarios, although the scenario number needed to build the response surface
for a specific variable number has already been dramatically reduced as compared to the
conventional RSM technique. Future studies are needed to further reduce the scenario number
and still retain the accuracy of the ERSM technique. Another disadvantage is that the current
ERSM technique does not consider the impact of meteorological variations on ambient
concentrations. We have detailed the advantages and disadvantages of the ERSM technique in
the revised manuscript (Page 4, Line 12-15; Page 8, Line 5-10; Page 21, Line 13-18).

ERSM could provide valuable information to develop effective strategies based on complex non-linear relationships. It means non-linear responses should represent the actual situation in the real atmosphere. I think validation of the responses obtained by ERSM is not enough whereas comparisons of observed concentrations have been made.

Especially, nonlinear responses of NOX emissions are critical for policy making. How much NOX reduction is necessary to realize positive effects to reduce PM2.5 concentrations? ERSM could give the answer. However, if the answer is not correct in the real atmosphere, policies may fail to realize PM2.5 reductions.

Response: We thank the reviewer for this valuable comment. We fully agree that the validation of the responses predicted by ERSM is very important. In response to the comment, we (1) strengthen the validation of the ERSM-predicted responses against CMAQ/2D-VBS simulation, (2) add some discussions about the evaluation of CMAQ/2D-VBS-simulated responses against the actual situation in the real atmosphere, and (3) add some discussions about the impact of $NO_X$ emission reductions.

(1) In the revised manuscript, we have added a group of scatter plots comparing the $PM_{2.5}$ responses (i.e., difference between $PM_{2.5}$ concentration in an emission control scenario and that in the base case) predicted by ERSM and independent CMAQ/2D-VBS simulations (second row of Fig. 2, shown below). Moreover, we have calculated the statistics for the comparison of $PM_{2.5}$ responses (Table 2, also shown below). Figure 2 and Table 2 illustrate that the ERSM-predicted and CMAQ/2D-VBS-simulated $PM_{2.5}$ responses agree well with each other. The correlation coefficients are larger than 0.99, and the normalized mean errors (NMEs) are within 5.6% for all four months. Note that we did not show the normalized errors (NEs) and mean normalized errors (MNEs) for $PM_{2.5}$ responses as we did for $PM_{2.5}$ concentrations in Table 2. The reason is that the CMAQ/2D-VBS-simulated $PM_{2.5}$ responses are very close to zero in several scenarios which are randomly generated, therefore their normalized errors (NEs) and mean normalized errors (MNEs) could be extremely large even if the absolute errors are small, which cannot properly characterize the accuracy of the ERSM technique.

In addition, we compare the 2D-isopleths of $PM_{2.5}$ concentrations as a function of continuous changes in precursor emissions (including $NO_X$ emissions) in a full range (from 0 to 1.2 times), derived from the ERSM and conventional RSM techniques (Fig. 3 in the manuscript). The predictions by conventional RSM can be regarded as proxies for real CMAQ/2D-VBS simulations since it has been extensively demonstrated to have high accuracy and stability in previous studies (Xing et al., 2011; Wang et al., 2011b). For this reason, the comparison between the ERSM and conventional RSM techniques helps to evaluate the accuracy and stability of the ERSM technique. The comparison shows that the shapes of isopleths derived from both prediction systems agree well with each other except for a few cases with very large emission reductions (> 80%), demonstrating the reliability of

ERSM in predicting the responses of $PM_{2.5}$ concentrations to changes in emissions of precursors, including $NO_X$. Note that all sensitivity scenarios used in the "Results and discussion" section have emission reductions ≤ 80%, therefore, the results and conclusions of this study are not affected by the relatively large errors at very large emission reductions.

(2) The preceding discussions demonstrate the agreement between ERSM-predicted and CMAQ/2D-VBS-simulated $PM_{2.5}$ responses. However, evaluating the $PM_{2.5}$ responses simulated by chemical transport models against the actual situation in the real atmosphere represents a major challenge in atmospheric modeling studies, because it is extremely difficult to artificially perturb emissions in the real atmosphere. Some special events when temporary control measures are implemented, such as the Beijing Olympic Games and the APEC conference, might provide opportunities to evaluate the simulated responses. However, such effects of temporary emission reductions could be confounded by meteorological variations. We fully recognize the importance to make sure that the simulated responses represent the situation in real atmosphere, but such evaluations are very complicated and appear to be beyond the purview of the present study. We have highlighted this issue as a major limitation of the present study (Page 21, Line 8-12), which requires further investigations.

[Figure]

Figure 2. Comparison of PM$_{2.5}$ concentrations (top row) and PM$_{2.5}$ responses (bottom row) predicted by the ERSM technique with out-of-sample CMAQ/2D-VBS simulations. The dashed line is the one-to-one line indicating perfect agreement.

Table 2. Comparison between ERSM-predicted and CMAQ/2D-VBS-simulated $PM_{2.5}$ concentrations for 54 out-of-sample scenarios.

| Month | Variable | Statistical index | Beijing | Tianjin | Northern Hebei | Eastern Hebei | Southern Hebei |
|-------|----------|-------------------|---------|---------|----------------|---------------|----------------|
| Jan | $PM_{2.5}$ concentration | R | 0.998 | 0.998 | 0.995 | 0.997 | 0.997 |
| | | MNE (%) | 0.52 | 0.55 | 0.64 | 0.67 | 0.60 |
| | | Maximum NE (%) | 7.56 | 6.98 | 10.67 | 8.01 | 8.03 |
| | | 95% percentile of NEs (%) | 1.61 | 2.86 | 2.92 | 3.46 | 3.02 |
| | | NME (%) | 0.44 | 0.46 | 0.57 | 0.53 | 0.53 |
| | $PM_{2.5}$ response | R | 0.998 | 0.998 | 0.995 | 0.997 | 0.997 |
| | | NME (%) | 3.36 | 3.48 | 4.25 | 4.00 | 3.88 |
| Mar | $PM_{2.5}$ concentration | R | 0.999 | 0.996 | 0.998 | 0.995 | 0.999 |
| | | MNE (%) | 0.37 | 0.54 | 0.39 | 0.57 | 0.49 |
| | | Maximum NE (%) | 3.75 | 6.58 | 4.30 | 5.04 | 3.22 |
| | | 95% percentile of NEs (%) | 1.53 | 3.15 | 2.03 | 4.35 | 2.03 |
| | | NME (%) | 0.31 | 0.45 | 0.34 | 0.49 | 0.42 |
| | $PM_{2.5}$ response | R | 0.999 | 0.996 | 0.998 | 0.995 | 0.999 |
| | | NME (%) | 2.38 | 4.32 | 2.70 | 4.55 | 3.59 |
| Jul | $PM_{2.5}$ concentration | R | 0.997 | 0.998 | 0.998 | 0.999 | 0.999 |
| | | MNE (%) | 0.94 | 0.54 | 0.46 | 0.37 | 0.47 |
| | | Maximum NE (%) | 5.05 | 5.02 | 4.65 | 1.83 | 3.62 |
| | | 95% percentile of NEs (%) | 3.47 | 2.33 | 2.17 | 1.49 | 1.87 |
| | | NME (%) | 0.80 | 0.47 | 0.41 | 0.33 | 0.39 |
| | $PM_{2.5}$ response | R | 0.997 | 0.998 | 0.998 | 0.999 | 0.999 |
| | | NME (%) | 4.97 | 3.71 | 2.80 | 2.58 | 2.78 |
| Oct | $PM_{2.5}$ concentration | R | 0.996 | 0.994 | 0.999 | 0.999 | 0.999 |
| | | MNE (%) | 0.83 | 0.70 | 0.36 | 0.39 | 0.36 |
| | | Maximum NE (%) | 8.90 | 11.19 | 3.79 | 3.90 | 2.46 |
| | | 95% percentile of NEs (%) | 3.04 | 3.50 | 1.44 | 2.10 | 1.64 |
| | | NME (%) | 0.67 | 0.58 | 0.30 | 0.35 | 0.32 |
| | $PM_{2.5}$ response | R | 0.996 | 0.994 | 0.999 | 0.999 | 0.999 |
| | | NME (%) | 4.51 | 5.64 | 2.20 | 3.29 | 2.79 |

(3) Next we discuss the impact of $NO_X$ emission reductions. If only the $NO_X$ emissions within the BTH region are controlled, our simulation results (Fig. 4) reveal that a very large reduction ratio (about 80%) is required to realize a reduction in annual $PM_{2.5}$ concentrations in most areas. However, the effects could be distinctly different if $NO_X$ emissions outside the BTH region are jointly reduced. Our previous studies using the CMAQ model (Zhao et al., 2013b; Wang et al., 2010; Wang et al., 2011b) have shown that uniform reductions in $NO_X$ emissions in the whole China by 23-50% result in considerable annual $PM_{2.5}$ reduction over the BTH region. This is because $NO_X$ emission reductions in upwind regions are more likely to result in a net

PM$_{2.5}$ decrease compared with local emission reductions, since the photochemistry typically changes from a NMVOC-limited regime in local urban areas at surface to a NO$_X$-limited regime in downwind areas or at upper levels (Xing et al., 2011). The simulation results in this paper also support the above-mentioned pattern and mechanism to some extent: even a 20% NO$_X$ emission reduction in BTH can lead to PM$_{2.5}$ decrease in Northern Hebei (see Fig. 4 in the manuscript), because, as the northernmost region in BTH, it is significantly affected by emissions in other regions within BTH. In view of the discussions above, we suggest that NO$_X$ emissions should be substantially reduced in the long run in both the BTH region and the other parts of China.

Finally, we note that NO$_X$ emissions were recently found to oxidize SO$_2$ in aerosol water, leading to additional PM$_{2.5}$ formation (Cheng et al., 2016; Wang et al., 2016), which is a missing chemical process in most chemical transport models. Incorporation of this process in the model may affect the simulated response of PM$_{2.5}$ to NO$_X$ emissions. More studies are still needed to further investigate the effects of NO$_X$ emissions on PM$_{2.5}$ concentrations. We have added the discussions above in the revised manuscript (from Page 10, Line 29 to Page 11, Line 3; Page 11, Line 10-32; Page 21, Line 18-23; from Page 13, Line 21 to Page 14, Line 6).

Which components are included in inorganic PM2.5? Is EC included? How about other components like metals? It looks strange that primary organic aerosol (POA) is included as a precursor probably due to treatment in VBS. Please give precise definitions of these words.

Response: We thank the reviewer for this valuable comment. Primary inorganic PM$_{2.5}$ is defined as all chemical components of primary PM$_{2.5}$ other than POA. By definition, it includes EC, metals, as well as many other constituents such as sulfate and nitrate directly emitted from sources. POA is treated as a precursor because it undergoes chemical reactions and produces SOA in the CMAQ/2D-VBS model, while primary inorganic PM$_{2.5}$ is chemically inert. In the revised manuscript, we have defined the "primary inorganic PM$_{2.5}$" clearly and added the reasons to treat POA as a precursor (from Page 8, Line 28 to Page 9, Line 1).

What do "discrepant temporary control strategies" mean? How are they possible? I understand major sources are different in each heavy air pollution episode. However, it could be possible to implement different temporary control strategies for each episode only if it could be forecasted. Can ERSM be used to forecast major sources in coming heavy air pollution episodes? I think differences of major sources in each episode suggest to implement strategies which control emissions of all the sources which could be major in various episodes.

Response: "Discrepant temporary control strategies" mean that the temporary control strategies should focus on different emission sources during different heavy pollution episodes. To make it clear, we have revised this sentence as follows:

The source contribution features for various types of heavy-pollution episodes are distinctly different from each other, and from the monthly mean results, illustrating that control strategies should be differentiated based on the major contributing sources during different types of episodes. (Page 2, Line 21-24)

In the present study, we only studied the source contribution features of three typical episodes. These results are not yet sufficient to guide the development of temporary control strategies for all heavy-pollution episodes, because the conclusions drawn from the three episodes may not be generalized to pollution types. In future studies, we need to simulate more episodes to improve their classification and to comprehensively understand the source contribution features of each pollution type. For a coming heavy-pollution episode, we can predict its pollution type using an air quality forecasting model, and subsequently formulate the temporary control strategies based on the source contribution features of this specific pollution type. We have described the method to develop episode-specific control strategies using ERSM in the revised manuscript (Page 19, Line 14-23).

Specific comments

Page 3, Line 8 How much are 2012 levels?

Response: It was not until January 2013 that the Ministry of Environment of China began to report $PM_{2.5}$ concentrations to the public. In 2012, the $PM_{2.5}$ concentrations were only available for limited sites such as the United States Embassy in Beijing, where the annual mean concentration was 90.7 $\mu g/m^3$. The average $PM_{2.5}$ concentrations over the BTH region were not publicly available.

Page 3, Line 22 The sentence here says "CTMs are the only feasible tools for evaluating the response of PM2.5 concentrations to emission changes". However, the sentences around the line 14 describe that embedding chemical tracers in chemical transport models (CTMs) cannot represent non-linear response. They may confuse some readers who are not familiar to CTMs.

Response: We agree with the reviewer and have deleted the former sentence, which is redundant.

Page 3, Line 26 "Sensitivities" are more appropriate than "contributions" in the context here.

Response: We agree with the reviewer and have modified this sentence as follows in the revised manuscript (Page 3, Line 25-31).

A number of studies have utilized the "Brute force" method to quantify the sensitivities of PM$_{2.5}$ concentrations over the BTH region to emissions from different spatial regions or different economic sectors, either on a seasonal basis or during a specific heavy-pollution episode.

Page 4, Line 5 How inadequate?

Response: The previous studies reviewed here applied the Decoupled Direct Method or Adjoint Analysis approach, which are used to calculate first-order sensitivities. However, characterizing the nonlinearity in the responses of PM$_{2.5}$ concentrations to emissions requires the calculation of second- or higher-order sensitivities. Therefore, we state that the previous studies have inadequately captured the nonlinearity in the responses of PM$_{2.5}$ concentrations to emissions. We have added the explanations to the revised manuscript (Page 4, Line 5-8).

Page 5, Line 7 How were emissions of IVOC provided?

Response: Following our previous study (Zhao et al., 2016), we assume IVOC emissions to be 30 times, 4.5 times, 1.5 times, and 3.0 times the POA emissions from gasoline vehicles, diesel vehicles, biomass burning, and other emission sources, respectively, which is based on a series of laboratory measurements (Gordon et al., 2014b; Gordon et al., 2014a; Hennigan et al., 2011; Jathar et al., 2014). We have added these descriptions in the revised manuscript (from Page 6, Line 32 to Page 7, Line 4).

Page 5, Line 8 OA and SOA are listed parallelly, but SOA is included in OA.

Response: We have revised the sentence as follows (Page 5, Line 6-11).

Compared with the default CMAQ, the CMAQ/2D-VBS model explicitly simulates aging of secondary organic aerosol (SOA) formed from non-methane volatile organic compounds (NMVOC), aging of primary organic aerosol (POA), and photo-oxidation of intermediate-volatility organic compounds (IVOC), thereby significantly improving the simulation results of organic aerosol (OA), particularly SOA.

Page 5, Line 30 I think NCEP final analysis data is not reanalysis data. Is it not used for grid nudging?

Response: We have revised the descriptions about the first guess field and nudging as follows in order to make them more accurate.

The National Center for Environmental Prediction (NCEP)'s FNL (Final) Operational Global Analysis data (ds083.2) at 1.0º × 1.0º and 6-h resolution are used to generate the first guess field. The NCEP's Automated Data Processing (ADP) data (ds351.0 and ds461.0) are used in objective analysis (i.e., grid nudging). (from Page 5, Line 31 to Page 6, Line 2 in the revised manuscript)

Page 6, Line 4 I think terrain data is not from MODIS.

Response: We apologize for the mistake and have corrected this sentence as follows:

The land cover type data are obtained from the Moderate resolution Imaging Spectroradiometer (MODIS). (Page 6, Line 7-8 in the revised manuscript)

Page 6, Line 25 How about open biomass burning emissions?

Response: In both the BTH and national emission inventories, the emissions from open burning of agricultural residue are calculated using crop yields, straw to grain ratio, fraction of biomass burned in the open field, and emission factors (Fu et al., 2013; Zhao et al., 2013a; Wang and Zhang, 2008). We do not include the emissions from forest and grassland fires, which typically account for less than 5% of the total biomass burning emissions over the BTH region (Qin and Xie, 2011) and are not the focus of the present study. We have added the preceding descriptions in the revised manuscript (Page 6, Line 23-28).

Page 9, Line 21 How about the performance of $SO_4^{2-}$, $NO_3^-$, and OA?

Response: This manuscript focuses on the response of $PM_{2.5}$ concentrations to air pollutant emissions, and the responses of $SO_4^{2-}$, $NO_3^-$, and OA are examined just to better understand the responses of $PM_{2.5}$. For this reason, the response surfaces of $SO_4^{2-}$, $NO_3^-$, and OA are only built using the conventional RSM technique to map their concentrations versus emissions of five $PM_{2.5}$ precursors, i.e., $NO_X$, $SO_2$, $NH_3$, NMVOC+IVOC, and POA. Since conventional RSM has been adequately demonstrated to have high accuracy and stability (Xing et al., 2011; Wang et al., 2011b), we did not include its validation in the present paper. We have clarified this point in the revised manuscript (Page 8, Line 18-22; Page 9, Line 27-29).

Page 10, Line 8 Why are only NMEs shown? How about R and MNEs? I suppose it is more important for RSM to see responses than to reproduce concentrations.

Response: We fully agree with the reviewer that the evaluation of $PM_{2.5}$ responses is very important. Since the CMAQ/2D-VBS-simulated $PM_{2.5}$ responses are very close to zero in several out-of-sample scenarios which are generated randomly, their normalized errors (NEs) and mean normalized errors (MNEs) could be extremely large even if the absolute errors are small, which cannot properly characterize the accuracy of the ERSM technique. For example, for the 11[th] case used in out-of-sample validation, the CMAQ/2D-VBS-simulated $PM_{2.5}$ response in January is 0.0003 $\mu g/m^3$ while the ERSM-predicted value is 0.03 $\mu g/m^3$. While the ERSM-predicted and CMAQ/2D-VBS-simulated values are actually quite close, the NE is as large as about 10000%. Therefore, we argue that NE and MNE are not suitable for evaluating ERSM's performance on $PM_{2.5}$ responses. With respect to R, the values for $PM_{2.5}$ responses are exactly the same as those for $PM_{2.5}$ concentrations, so we did not include R for $PM_{2.5}$ responses in the original manuscript. In the revised manuscript, we have added R for $PM_{2.5}$ responses to make the results more clear (Table 2), and also explained the reasons for excluding NE and MNE (from Page 10, Line 30 to Page 11, Line 1).

Page 10, Line 15 I do not understand meaning of comparisons between ERSM and conventional RSM. Why these two model could produce different results? Which should be correct? The sentence in the line 31 says that the ERSM predictions are definitely subject to numerical errors, but I do not know why "definitely". Although there are descriptions of ERSM in the first paragraph of the section 2.2, the advantages and disadvantages of ERSM against conventional RSM should be clearly explained.

Response: We thank the reviewer for this valuable comment. While the conventional RSM has been demonstrated to have very high accuracy and stability, the number of emission scenarios required to build it depends on the variable number via an equation of fourth or higher order. Therefore, the required scenario number would be tens of thousands for over 15 variables and even hundreds of thousands for over 25 variables, which is computationally impossible for most three-dimensional CTMs and proves to be a major limitation for the conventional RSM technique. The ERSM technique substantially reduces the number of scenarios needed to build the response surface by introducing several additional assumptions with respect to the inter-regional transport of air pollutants (see Section 2.2), which extends its applicability to a much larger number of regions, pollutants, and sectors with an acceptable computational burden. Meanwhile, the additional assumptions in the ERSM techniques might affect its accuracy. Therefore, the conventional RSM technique is theoretically more close to the predictions of CMAQ/2D-VBS, and its accuracy has been extensively evaluated in previous studies (Xing et al., 2011; Wang et al., 2011b). For this reason, the comparison between the ERSM and conventional RSM techniques helps to evaluate the accuracy and stability of the ERSM technique.

The statement that "ERSM predictions are definitely subject to numerical errors" means that ERSM, like all models, cannot exactly agree with the true values. We have deleted this redundant sentence in the revised manuscript to avoid misunderstanding.

As described above, the major advantage of ERSM over conventional RSM is that it is applicable to a much larger number of regions, pollutants, and sectors with an acceptable computational burden. For example, in the present study, the conventional RSM is applied to only 5 control variables, i.e., the total emissions of five $PM_{2.5}$ precursors. The ERSM technique, however, is applied to 55 control variables including the emissions of multiple pollutants from different regions and sectors. The major disadvantage of ERSM is that it might be subject to larger errors than conventional RSM due to the additional assumptions in the treatment of inter-regional transport.

We have added the descriptions above in the revised manuscript (Page 4, Line 19-27; Page 11, Line 10-17).

Page 11, Line 5 What is the advantage of ERSM against conventional RSM in the results shown in Figure 4? I think the sector-wise results shown in the right figure cannot be obtained by conventional RSM. Is that correct? Please described what is newly obtained by using ERSM.

Response: It is correct. The sector-wise results shown in Fig. 4 (right panel) and Fig. 5, as well as the regional contributions shown in Fig. 6 can only be obtained from ERSM. We have clarified this in the revised manuscript (Page 12, Line 15-17).

Page 11, Line 16 It looks strange to represent primary inorganic PM2.5 as "single pollutant" because it is a mixture of various components in fact.

Response: We have modified this sentence as follows:

While primary inorganic $PM_{2.5}$ makes the largest contribution to $PM_{2.5}$ concentrations among all air pollutants, the total contributions of all precursors ($NO_X$, $SO_2$, $NH_3$, NMVOC, IVOC, and POA), which range between 31% and 48%, exceed that of primary inorganic $PM_{2.5}$ (24-36%). (Page 12, Line 25-28)

Page 11, Line 29 What is the reasons of small sensitivities of SO2 emissions on PM2.5?

Response: In the manuscript, we state that the $PM_{2.5}$ sensitivity to $SO_2$ emissions is smaller than that to POA, NMVOC+IVOC, and $NH_3$. From 2007 to 2014 (the base year of this study), both $SO_2$ emissions and $SO_4^{2-}$ concentrations in $PM_{2.5}$ have been continuously decreasing due to effective control policies (Wang et al., 2017). As a result, the simulated concentrations of $SO_4^{2-}$ are much lower than those of OA (see Fig. 7 and Fig. S7 in the manuscript), which explains the smaller sensitivity of $PM_{2.5}$ to $SO_2$ than those to POA and NMVOC+IVOC. The reason why $PM_{2.5}$ is less sensitive to $SO_2$ emission reductions than that to $NH_3$ is that the reduction in $NH_3$ emissions affects both the concentrations of $NO_3^-$ and $SO_4^{2-}$, while $SO_2$ emission reductions mainly lead to decrease in $SO_4^{2-}$ concentrations. Additionally, the small sensitivities to $SO_2$ emissions may also be partly attributed to the underestimation of $SO_4^{2-}$ in the CMAQ/2D-VBS model, which is a common problem of many chemical transport models (Wang et al., 2011a; Gao et al., 2014; Wang et al., 2013). While the reasons for underestimation are yet to be resolved, possible causes could be the lack of some chemical formation pathways in the modeling system, such as $SO_2$ heterogeneous reactions on the dust surface and the oxidation of $SO_2$ by $NO_2$ in aerosol water (Wang et al., 2013; Fu et al., 2016; Cheng et al., 2016). We have added the discussions in the revised manuscript (Page 13, Line 8-11; Page 14, Line 2-6).

Page 11, Line 31 Nonlinear sensitivities of NOX emissions and their changes from negative to positive are described from here. I also agree that this is very important phenomena to consider effective emission controls. However, on the other hand, the descriptions in the page 10 treat such a nonlinear change in sensitivities and differences with conventional RSM as just a rare case involving large unrealistic reduction of NOx emissions. I do not agree that. Even if large NOx reduction is required, the performance of ERSM to represent such a nonlinear change should be carefully evaluated.

Response: We agree with the reviewer that we should carefully evaluate the performance of ERSM over a full emission range, including at very large $NO_X$ emission reductions. The reason why we stated that the relatively large errors at very low emission ratios did not affect our conclusion is that all sensitivity scenarios used in the "Results and discussion" section have emission ratios $\geq$ 0.2. In response to the reviewer's comment, we have strengthened the validation of ERSM-predicted $PM_{2.5}$ responses against CMAQ/2D-VBS simulations, as described in detail in our response to the reviewer's second "general comment". On the other hand, we have added a detailed discussion about the relatively large errors at very low $NO_X/NH_3$ emission ratios (< 0.2), and highlighted the need for further studies (from Page 11, Line 24 to Page 12, Line 6). The revised text is shown below.

The agreement is very good for the case of VOC+IVOC vs POA, and for the cases of $NO_X$ vs $NH_3$ and $SO_2$ vs $NH_3$ when the emission ratios for $NO_X$ and $NH_3$ are larger than 0.2. Relatively large errors occur at very low $NO_X/NH_3$ emission ratios (< 0.2) due primarily to an extremely strong nonlinearity. Within these low emission ranges, the ERSM technique can capture the general trends in $PM_{2.5}$ concentrations in response to emission changes, but the concentration gradients predicted by ERSM are smaller than those given by conventional RSM. More studies are needed to further improve the performance of ERSM at very low $NO_X/NH_3$ emission ratios.

Finally, we note that all sensitivity scenarios used in the "Results and discussion" section have emission ratios ≥ 0.2, therefore, the results and conclusions of this study are not affected by the relatively large errors at very low $NO_X/NH_3$ emission ratios.

Page 12, Line 2 Indeed, the regimes are very important for negative and positive sensitivities of NOX emissions. Therefore, it is quite important to see if ERSM could accurately represent regimes in the real atmosphere. I suppose such validations are scarce.

Response: Although the ERSM-predicted responses of $PM_{2.5}$ concentrations have been demonstrated to agree fairly well with CMAQ/2D-VBS simulations, evaluating the simulated $PM_{2.5}$ responses (or chemical regimes) against the actual situation in the real atmosphere represents a major challenge in atmospheric modeling studies, because it is extremely difficult to artificially perturb emissions in the real atmosphere. Some special events when temporary control measures are implemented, such as the Beijing Olympic Games and the APEC conference, might provide opportunities to evaluate the simulated responses. However, such effects of temporary emission reductions could be confounded by meteorological variations. We fully recognize the importance to make sure that the simulated responses represent the situation in real atmosphere, but such evaluations are very complicated and appear to be beyond the purview of the present study. We have highlighted this issue as a major limitation of the present study (Page 21, Line 18-23), which requires further investigations.

Page 12, Line 20 Are there any discussions on differences between sensitivities of all pollutants and sectors and sum of sensitivities of individual pollutants and sectors?

Response: The sum of sensitivities of $PM_{2.5}$ concentrations to individual pollutant-sector combinations is mostly larger than the sensitivity to all pollutants and sectors, especially under large reduction ratios. This is mainly attributed to the overlapping effect of two precursors (e.g., $SO_2$ and $NH_3$) involved in the formation of ammonium sulfate and ammonium nitrate. Nevertheless, at small reduction ratios, the sum of individual sensitivities is sometimes smaller, because the negative effects of reducing $NO_X$ are mitigated when we simultaneously reduce $NO_X$ emissions from multiple sectors as well as emissions of other air pollutants such as NMVOC. We have included these discussions in the revised manuscript (Page 14, Line 11-18).

Page 12, Line 31 What is a reason of higher sensitivities of residential and commercial sources in winter? Heating?

Response: There are two major reasons. On one hand, as the reviewer points out, the emissions from residential and commercial sources are relatively higher in winter due to heating. On the other hand, the weaker vertical mixing in winter also results in a larger relative contribution of low-level sources including the residential and commercial sector. We have added these explanations in the revised manuscript (Page 14, Line 29-32).

Page 13, Line 8 Are there any specific results indicating the importance of NOx emissions outside the BTH region?

Response: The present study focuses on the response of $PM_{2.5}$ concentrations to emissions within the BTH region. If only the $NO_X$ emissions within the BTH region are controlled, a very large reduction ratio of about 80% is required to realize a reduction in annual $PM_{2.5}$ concentrations in most areas (Fig. 4). However, our previous studies using the CMAQ model (Zhao et al., 2013b; Wang et al., 2010; Wang et al., 2011b) have shown that uniform reductions in $NO_X$ emissions in the whole China by 23-50% result in considerable annual $PM_{2.5}$ reduction over the BTH region, implying the important role of $NO_X$ emission reductions outside the BTH region. The reason why $NO_X$ emission reductions in upwind regions are more likely to result in a net $PM_{2.5}$ decrease compared with local emission reductions is that the photochemistry typically changes from a NMVOC-limited regime in local urban areas at surface to a $NO_X$-limited regime in downwind areas or at upper levels (Xing et al., 2011). The simulation results in this paper also support the above-mentioned pattern and mechanism to some extent: even a 20% $NO_X$ emission reduction in BTH can lead to $PM_{2.5}$ decrease in Northern Hebei (see Fig. 4 in the manuscript), because, as the northernmost region in BTH, it is significantly affected by emissions in other regions within BTH. We have added these discussions in the revised manuscript (from Page 13, Line 21 to Page 14, Line 2).

Page 14, Line 6 How does seasonal variations of NH3 emissions look like?

Response: The monthly variations in $NH_3$ emissions from fertilizer application are based on our previous simulation results (Fu et al., 2015) using an agricultural fertilizer modeling system which couples a regional air quality model (the Community Multi-scale Air Quality model, or CMAQ) and an agro-ecosystem model (the Environmental Policy Integrated Climate model, or EPIC). The monthly variations of livestock farming are obtained from Huang et al. (2012), and those of other emission sources are consistent with the descriptions in our previous paper (Wang et al., 2011a). Overall, the monthly variations in total $NH_3$ emissions are illustrated in the following figure.

[Figure]

Figure. Monthly variations in total NH$_3$ emissions over the BTH region.

Page 14, Line 25 Is it confirmed that NOX competes with SO2 for NH3 in a thermodynamic pathway? I think SO42- is much more predominantly in aerosol phase than NO3-.

Response: We agree that NH$_3$ tends to react with SO$_2$ to form ammonium sulfate. In the present study, the response of SO$_4^{2-}$ concentrations to NO$_X$ emissions can be well explained by only the changes in O$_3$ and HO$_X$ concentrations, i.e., the photochemical pathway. Also, the BTH region has been shown to be generally under an NH$_3$-rich condition (Wang et al., 2011b). Therefore, the competition between NO$_X$ and SO$_2$ for NH$_3$ does not appear to play a noticeable role in changing SO$_4^{2-}$ concentrations. In the revised manuscript, we have deleted the descriptions about the thermodynamic pathway and focused on the photochemical pathway.

Page 15, Line 1 Does this POA include semivolatile components which could condensate only under lower temperature in winter?

Response: We agree with the reviewer that POA includes some semi-volatile components which tend to partition to the particle phase under low temperature in January, which partly explains the higher contributions of POA emissions to OA concentrations in January. Besides, some other factors account for the higher contributions of POA emissions in January and higher contributions of NMVOC+IVOC emissions in July. First, the POA emissions are relatively higher in January due to residential heating, while the NMVOC emissions from solvent use and biogenic sources are higher in July. Second, higher temperature and stronger radiation in July accelerate the formation of SOA from NMVOC+IVOC. We have added the explanations in the revised manuscript (Page 17, Line 3-9).

Page 17, Line 10 I agree more model simulations of more episodes are necessary, but a model can always give results. I believe what is important is to confirm model results are consistent with actual situations in the real atmosphere. That is quite important to consider effective strategies for heavy air pollutions.

Response: We appreciate the reviewer's valuable comment. We have discussed the validation of $PM_{2.5}$ responses predicted by ERSM in detail in our response to the reviewer's second "general comment". In brief, we strengthened the validation of ERSM-predicted responses against CMAQ/2D-VBS simulations and have demonstrated that the ERSM-predicted and CMAQ/2D-VBS-simulated responses of $PM_{2.5}$ concentrations to precursor emissions, including $NO_X$ emissions, agree fairly well with each other. However, evaluating the $PM_{2.5}$ responses simulated by CMAQ/2D-VBS against the actual situation in the real atmosphere represents a major challenge in atmospheric modeling studies, because it is extremely difficult to artificially perturb emissions in the real atmosphere. We have recognized this issue as a major limitation of the present study, which requires further investigations.

Page 18, Line 18 I am wondering if NMVOC and IVOC should be discussed together to implement any strategies because their sources and their effects on PM2.5 and ozone could be different.

Response: We fully agree with the reviewer that the impact of NMVOC and IVOC emissions should ideally be quantified separately considering the differences in their sources and effects on SOA and $O_3$. In the present study, they are lumped together to reduce the number of control variables in view of the fact that they have many common sources and could be controlled using similar removal technologies. To better inform NMVOC/IVOC control policies, it is needed in future studies to perform a detailed quantification of the individual contributions of NMVOC and IVOC emissions from various sources to $PM_{2.5}$ concentrations. We have described this limitation at the end of the revised manuscript (Page 21, Line 23-27).

Page 18, Line 24 I agree NOx reduction is necessary in the long run. However, it could increase PM2.5 emissions in the near term with slight reduction. How should such adverse effects be considered? Any messages on this issue?

Response: We suggest that, in the long run, $NO_X$ emissions should be substantially reduced, preferably approach their maximum feasible reduction levels, in both the BTH and other parts of China. In the short term, we should also implement simultaneous $NO_X$ reductions in both the BTH and other regions in order to avoid the adverse effects. We have added this suggestion to the revised manuscript (Page 21, Line 4-6).

Page 18, Line 26 I feel the importance of Southern Hebei is not so discussed in the main text.

Response: We have better discussed the importance of Southern Hebei in the revised manuscript (Page 15, Line 21-27):

The precursor emissions from the northern part of BTH (e.g., Northern Hebei, Beijing) mainly contribute to local $PM_{2.5}$ concentrations, whereas those from the southern part of BTH (e.g., Southern Hebei) significantly affect the $PM_{2.5}$ concentrations in both the local region and other regions. Over the BTH, heavy pollution is frequently associated with southerly wind while strong northerly wind often blows away $PM_{2.5}$ pollution (Jia et al., 2008; Zheng et al., 2015), which explains the higher contribution of emissions from southern BTH to other regions.

Technical corrections

Page 6, Line 17 origninally -> originally

Response: Revision has been made.

Table 2. Comparison between ERSM-predicted and CMAQ/2D-VBS-simulated $PM_{2.5}$ concentrations for

54 out-of-sample scenarios.

| Month | Variable | Statistical index | Beijing | Tianjin | Northern Hebei | Eastern Hebei | Southern Hebei |
|---|---|---|---|---|---|---|---|
| Jan | $PM_{2.5}$ concentration | R | 0.998 | 0.998 | 0.995 | 0.997 | 0.997 |
| | | MNE (%) | 0.52 | 0.55 | 0.64 | 0.67 | 0.60 |
| | | Maximum NE (%) | 7.56 | 6.98 | 10.67 | 8.01 | 8.03 |
| | | 95% percentile of NEs (%) | 1.61 | 2.86 | 2.92 | 3.46 | 3.02 |
| | | NME (%) | 0.44 | 0.46 | 0.57 | 0.53 | 0.53 |
| | $PM_{2.5}$ response | R | 0.998 | 0.998 | 0.995 | 0.997 | 0.997 |
| | | NME (%) | 3.36 | 3.48 | 4.25 | 4.00 | 3.88 |
| Mar | $PM_{2.5}$ concentration | R | 0.999 | 0.996 | 0.998 | 0.995 | 0.999 |
| | | MNE (%) | 0.37 | 0.54 | 0.39 | 0.57 | 0.49 |
| | | Maximum NE (%) | 3.75 | 6.58 | 4.30 | 5.04 | 3.22 |
| | | 95% percentile of NEs (%) | 1.53 | 3.15 | 2.03 | 4.35 | 2.03 |
| | | NME (%) | 0.31 | 0.45 | 0.34 | 0.49 | 0.42 |
| | $PM_{2.5}$ response | R | 0.999 | 0.996 | 0.998 | 0.995 | 0.999 |
| | | NME (%) | 2.38 | 4.32 | 2.70 | 4.55 | 3.59 |
| Jul | $PM_{2.5}$ concentration | R | 0.997 | 0.998 | 0.998 | 0.999 | 0.999 |
| | | MNE (%) | 0.94 | 0.54 | 0.46 | 0.37 | 0.47 |
| | | Maximum NE (%) | 5.05 | 5.02 | 4.65 | 1.83 | 3.62 |
| | | 95% percentile of NEs (%) | 3.47 | 2.33 | 2.17 | 1.49 | 1.87 |
| | | NME (%) | 0.80 | 0.47 | 0.41 | 0.33 | 0.39 |
| | $PM_{2.5}$ response | R | 0.997 | 0.998 | 0.998 | 0.999 | 0.999 |
| | | NME (%) | 4.97 | 3.71 | 2.80 | 2.58 | 2.78 |
| Oct | $PM_{2.5}$ concentration | R | 0.996 | 0.994 | 0.999 | 0.999 | 0.999 |
| | | MNE (%) | 0.83 | 0.70 | 0.36 | 0.39 | 0.36 |
| | | Maximum NE (%) | 8.90 | 11.19 | 3.79 | 3.90 | 2.46 |
| | | 95% percentile of NEs (%) | 3.04 | 3.50 | 1.44 | 2.10 | 1.64 |
| | | NME (%) | 0.67 | 0.58 | 0.30 | 0.35 | 0.32 |
| | $PM_{2.5}$ response | R | 0.996 | 0.994 | 0.999 | 0.999 | 0.999 |
| | | NME (%) | 4.51 | 5.64 | 2.20 | 3.29 | 2.79 |

[Figure]

Figure 1. Double nesting domains used in CMAQ/2D-VBS simulation (left) and the definition of five target regions in the innermost domain, denoted by different colours (right). The grey lines in the right figure represent the boundaries of prefecture-level cities.

[Figure]

Figure 2. Comparison of PM$_{2.5}$ concentrations (top row) and PM$_{2.5}$ responses (bottom row) predicted by the ERSM technique with out-of- sample CMAQ/2D-VBS simulations. The dashed line is the one-to-one line indicating perfect agreement.

[Figure]

Figure 3. Comparison of the 2-D isopleths of PM$_{2.5}$ concentrations in Beijing in response to the simultaneous changes of precursor
emissions in all five regions derived from the conventional RSM technique and the ERSM technique. The X- and Y-axis represent the
emission ratio, defined as the ratios of the changed emissions to the emissions in the base case. The colour contours represent PM$_{2.5}$
concentrations (unit: μg m$^{-3}$).

[Figure]

[Figure]

Figure 3. Continued.

[Figure]

Figure 4. Sensitivity of 4-month mean PM$_{2.5}$ concentrations to stepped control of individual air pollutants (left) and individual pollutant-sector combinations (right). The X-axis shows the reduction ratio (= 1 – emission ratio). The Y-axis shows PM$_{2.5}$ sensitivity, which is defined as the change ratio of concentration divided by the reduction ratio of emissions. The coloured bars denote the PM$_{2.5}$ sensitivities when a particular emission source is controlled while the others stay the same as the base case; the black dotted line denotes the PM$_{2.5}$ sensitivity when all emission sources are controlled simultaneously. The red stars represent PM$_{2.5}$ concentrations in the base case.

[Figure]

Figure 5. Sensitivity of monthly mean PM$_{2.5}$ concentrations to stepped control of individual air pollutants from individual sectors in January, March, July, and October. The meanings of

X-axis, Y-axis, coloured bars, black dotted lines, and red stars are the same as Fig. 4.

[Figure]

Figure 6. Contributions of precursor (NO$_X$, SO$_2$, NH$_3$, NMVOC, IVOC, and POA) and primary inorganic PM$_{2.5}$ emissions from individual regions to PM$_{2.5}$ concentrations. The contributions are quantified by comparing the base case with sensitivity scenarios in which emissions from a specific source are reduced by 80%. This figure illustrates contributions to

4-month mean PM$_{2.5}$ concentrations and monthly mean PM$_{2.5}$ concentrations in January and

July. The results for March and October are given in Fig. S6.

[Figure]

Figure 7. Sensitivity of monthly mean $NO_3^-$, $SO_4^{2-}$, and OA concentrations to stepped control of individual air pollutants in January and July. The meanings of X-axis, Y-axis, coloured bars, black dotted lines, and red stars are the same as Fig. 4 but for $NO_3^-/SO_4^{2-}/OA$. The results for March and October are given in Fig. S7.

[Figure]

Figure 8. Contribution of precursor (NO$_X$, SO$_2$, NH$_3$, NMVOC, IVOC, and POA) and primary inorganic PM$_{2.5}$ emissions from individual regions to PM$_{2.5}$ concentrations during three typical heavy-pollution episodes.

[Figure]

Figure 9. Sensitivity of PM$_{2.5}$ concentrations to stepped control of individual air pollutants from individual sectors during three heavy-pollution episodes. The meanings of X-axis, Y-axis, coloured bars, and black dotted lines are the same as Fig. 4.